# MultiCoNER v2: a Large Multilingual dataset for Fine-grained and Noisy Named Entity Recognition

**Besnik Fetahu    Zhiyu Chen    Sudipta Kar    Oleg Rokhlenko    Shervin Malmasi**

Amazon.com, Inc.    Seattle, WA, USA

{besnikf,zhiyuche,sudipkar,olegro,malmasi}@amazon.com

## Abstract

We present MULTICONER V2, a dataset for fine-grained Named Entity Recognition covering 33 entity classes across 12 languages, in both monolingual and multilingual settings. This dataset aims to tackle the following practical challenges in NER: (i) effective handling of fine-grained classes that include complex entities like movie titles, and (ii) performance degradation due to noise generated from typing mistakes or OCR errors.

The dataset is compiled from open resources like Wikipedia and Wikidata, and is publicly available.[1] Evaluation based on the XLM-RoBERTa baseline highlights the unique challenges posed by MULTICONER V2: (i) the fine-grained taxonomy is challenging, where the scores are low with macro-F1=0.63 (across all languages), and (ii) the corruption strategy significantly impairs performance, with entity corruption resulting in 9% lower performance relative to non-entity corruptions across all languages. This highlights the greater impact of entity noise in contrast to context noise.

## 1 Introduction

Named Entity Recognition (NER) is a core task in Natural Language Processing that involves identifying entities and recognizing their type (e.g., *person* or *location*). Recently, Transformer-based NER approaches have achieved new state-of-the-art (SOTA) results on well-known benchmark datasets like CoNLL03 and OntoNotes (Devlin et al., 2019). Despite these strong results, as shown by Malmasi et al. (2022a), there remain a number of practical challenges that are not well represented by existing datasets, including low context, diverse domains, missing casing information, and text with grammatical and typographical errors. Furthermore, wide use of coarse-grained taxonomies often contribute to a lower utility of NER systems, as additional fine-grained disam-

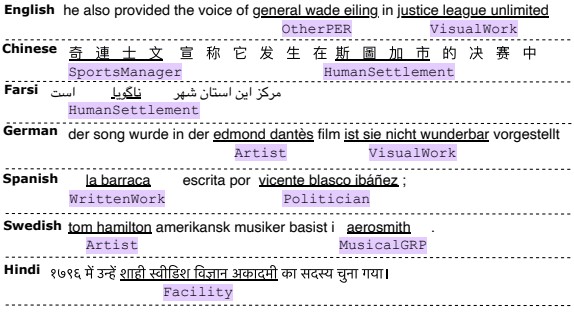

Figure 1: Example sentences from MULTICONER V2.

biguation steps are needed in many practical systems. Furthermore, models trained on datasets such as CoNLL03 perform significantly worse on unseen entities or noisy texts (Meng et al., 2021).

### 1.1 Real-world Challenges in NER

Outside the news domain used in datasets like CoNLL03, there are many challenges for NER. We categorize the challenges (cf. Table 5 in Appendix A) typically encountered in NER by several dimensions: (i) available context around entities, (ii) named entity surface form complexity, (iii) frequency distribution of named entity types, (iv) multilinguality, (v) fine-grained entity types, and (vi) noisy text containing typing errors. MULTICONER V2 represents all of these challenges, with a focus on (v) and (vi).

**Fine-Grained Entity Types** Most datasets use coarse types (e.g., PERSON, LOCATION in CoNLL03), with each type subsuming a wide range of entities with different properties. For instance, for LOCATION, different entities can appear in different contexts and have diverging surface form token distributions, which can lead to challenges in terms of NER, e.g., *airports* as part of LOCATION are often named after notable persons, thus for sentences with low context, NER models can confuse them with PERSON entities.

We introduce a fine-grained named entity taxonomy with 33 classes, where coarse grained types are further broken down into fine-grained types.

---
[1] https://registry.opendata.aws/multiconer
https://huggingface.co/datasets/MultiCoNER/multiconer_v2
https://www.kaggle.com/datasets/cryptexcode/multiconer-2

Additionally, we introduce types from the medical domain, presenting a further challenge for NER. Fine-grained NER poses challenges like distinguishing between different fine-grained types of a coarse type. For example, correctly identifying fine-grained types (e.g., SCIENTIST, ARTIST) of a PERSON entity requires effective representation of the *context* and often external world knowledge. Since two instances belonging to different types may have the same name, the context is needed to correctly disambiguate the types.

**Noisy Text**   Existing NER datasets mostly consist of well-formed sentences. This creates a gap in terms of training and evaluation, and datasets assume that models are applied only on well-formed content. However, NER models are typically used in web settings, exposed to user-created content. We present a noisy test set in MULTICONER V2, representing typing errors affected by keyboard layouts. This noise, impacting named entity and context tokens, presents challenges for accurately identifying entity spans.

**Contributions**   Our contributions through this work can be summarized as follows:

1. We created a taxonomy of 33 fine-grained NER classes across six coarse categories (PERSON, LOCATION, GROUP, PRODUCT, CREATIVEWORK, and MEDICAL). We use the taxonomy to develop a dataset for 12 languages based on this taxonomy. Our experiments show that NER models have a large performance gap ($\approx$14% macro-F1) in identifying fine-grained classes versus coarse ones.

2. We implement techniques to mimic web text errors, such as typos, to enhance and test NER model resilience to noise.

3. Experimentally we show that noise can significantly impact NER performance. On average, across all languages, the macro-F1 score for corrupted entities is approximately 9% lower than that for corrupted context, emphasizing the greater influence of entity noise as opposed to context noise.

MULTICONER V2 has been used as the main dataset for the MULTICONER V2 shared task (Fetahu et al., 2023) at SemEval-2023.

## 2   MULTICONER V2 Dataset Overview

MULTICONER V2 was designed to address the challenges described in §1.1. It contains 12 languages, including multilingual subsets, with a total of 2.3M instances with 32M tokens and a total of 2.2M entities (1M unique). While focused on different challenges, we adopted the data construction process described of the original MULTICONER (v1) corpus Malmasi et al. (2022a) and details about the process are provided in Appendix B. While both the v1 and v2 datasets focus on complex entities, MULTICONER V2 introduces a fine-grained taxonomy, adds realistic noise as well as 5 new languages.

Next we describe the fine-grained NER taxonomy construction and noise generation processes.

### 2.1   NER Taxonomy

MULTICONER V2 builds on top of the WNUT 2017 (Derczynski et al., 2017) taxonomy entity types, with the difference that it groups GROUP and CORPORATION types into one,[2] and introduces the MEDICAL type. Next, we divide coarse classes into fine-grained sub-types. Table 1 shows the 33 fine-grained classes, grouped across 6 coarse grained types. Our taxonomy is inspired by DBpedia's ontology (Auer et al., 2007), where a subset of types are chosen as fine-grained types that have support in encyclopedic resources.

This taxonomy allows us to capture fine-grained differences in entity types, where the entity context plays a crucial role in distinguishing the different types, as well as cases where external knowledge can help where context may be insufficient (e.g.,PRIVATECORP vs. PUBLICCORP).

### 2.2   Languages and Subsets

MULTICONER V2 includes 12 languages: Bangla (BN), Chinese (ZH), English (EN), Farsi (FA), French (FR), German (DE), Hindi (HI), Italian (IT), Portuguese (PT), Spanish (ES), Swedish (SV), Ukrainian (UK). The chosen languages span diverse typology, writing systems, and range from well-resourced (e.g., EN) to low-resourced (e.g., FA, BN).

**Monolingual Subsets**   Each of the 12 languages has their own subset with data from all domains, consisting of a total of 2.3M instances.

---

[2]The original definition is ambiguous in the difference between a CORPORATION and GROUP entity.

| PER (Person) | LOC (Location) | GRP (Group) | PROD (Product) | CW (Creative Work) | MED (Medical) |
|---|---|---|---|---|---|
| ARTIST | FACILITY | AEROSPACEMANUFACTURER | CLOTHING | ARTWORK | ANATOMICALSTRUCTURE |
| ATHLETE | HUMANSETTLEMENT | CARMANUFACTURER | DRINK | MUSICALWORK | DISEASE |
| CLERIC | STATION | MUSICALGRP | FOOD | SOFTWARE | MEDICALPROCEDURE |
| POLITICIAN | OTHERLOC | ORG | VEHICLE | VISUALWORK | MEDICATION/VACCINE |
| SCIENTIST | | PRIVATECORP | OTHERPROD | WRITTENWORK | SYMPTOM |
| SPORTSMANAGER | | PUBLICCORP | | | |
| OTHERPER | | SPORTSGRP | | | |

Table 1: MULTICONER V2 fine-grained NER taxonomy. There are 33 fine-grained classes, grouped across 6 coarse types.

**Multilingual Subset**   This subset is designed for evaluation of multilingual models, and should be used under the assumption that the language for each sentence is unknown. The training and development set from the monolingual splits are merged, while, for the test set, we randomly select at most 35,000 samples resulting in a total of 358,668 instances. The test set consists of a subset of samples that incorporate simulated noise.

## 2.3   Noise Generation

To emulate realistic noisy scenarios for NER, we developed different corruption strategies that reflect typing errors by users. We considered two main strategies: (i) replace characters in a token with neighboring characters (mimicking typing errors) on a given keyboard layout for a target language, and (ii) replace characters with visually similar characters (mimicking incorrectly visually recognized characters, e.g., OCR).

We applied the noise only on the test set[3] for a subset of languages (EN, ZH, IT, ES, FR, PT and SV). The noise can corrupt *context* tokens and *entity* tokens. For ZH, the corruption strategies are applied at the character level. We followed Wang et al. (2018)'s approach that automatically extracts visually similar characters using OCR tools and phonologically similar characters using ASR tools. We used their constructed dictionary[4] to match targeting characters.

For the other six languages, we developed word-level corruption strategies based on common typing mistakes, utilizing language specific keyboard layouts.[5] Table 2 summarizes the designed corruptions and their probabilities of occurrence.

**Noise Sampling**   We introduce synthetic noise into 30% of the test set. For each corrupted sentence, the number of corrupted tokens $\delta$ is sampled

| Corruption Strategy | Prob. |
|---|---|
| Adds a neighbor letter on keyboard next to a random token letter. | 0.1 |
| Replaces a random word letter with its keyboard-neighbor letter. | 0.2 |
| Replaces a random word letter with another visually similar one. | 0.2 |
| Skips a random word letter. | 0.2 |
| Swaps two random consecutive word letters. | 0.1 |
| Repeats a random word letter. | 0.2 |

Table 2: Corruption strategies and the probability with which they are applied.

from a truncated Poisson distribution ($\lambda = 1$ and $\delta \leq 3$). Since the minimum of $\delta$ can be zero, we use $\delta' = \delta + 1$ as the final corruption count.

Corruptions can apply to entity or non-entity (context) tokens. For ZH, we corrupt a context or entity character with 15% and 85% probability respectively. Using the corruption dictionary, we replace the original character at random with a visually or phonologically similar character from identified candidates. For the other languages, the probability of corrupting entity tokens is set to 80%. This is lower given that each letter in entities can be corrupted and the number of candidates in entities is more than that in ZH. On a word chosen for corruption, we perform a corruption action based on probabilities shown in Table 2. Examples of corrupted sentences are shown in Figure 2.

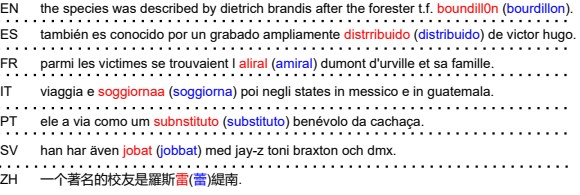

Figure 2: Examples of corrupted sentences. The red tokens are the corrupted versions of the blue tokens.

[3]To emulate a realistic scenario, where models are trained on clean text, but are exposed directly to user input.
[4]https://github.com/wdimmy/Automatic-Corpus-Generation
[5]We extended the keyboard layouts to include 7 languages: https://github.com/ranvijaykumar/typo.

# 3 NER Model Performance

We assess if MULTICONER V2 is challenging (cf. §1.1) by training and testing with XLM-RoBERTa (Conneau et al., 2020, XLMR).

**Metrics** We measure the model performance using Precision (P), Recall (R), and F1, where for F1 we distinguish between *micro/macro* averages.

## 3.1 Results

Table 3 shows the results obtained for the XLMR baseline. The results show the micro-F1 scores achieved on the individual NER coarse classes (See Appendix C for fine-grained performance).

Across all subsets, XLMR achieves the highest performance of micro-F1=0.77 for HI, and lowest micro-F1=0.61 for EN, ES and FA. The reason for the higher F1 scores on HI (or BN second highest score) is due to the smaller test set size, since the data is translated and is more scarce. For the rest, the F1 scores are typically lower, this is is mainly associated with the larger test sizes.

| Lang. | Macro-F1 | Micro-F1 | Lang. | Macro-F1 | Micro-F1 |
|-------|----------|----------|-------|----------|----------|
| BN | 0.68 | 0.74 | IT | 0.58 | 0.63 |
| DE | 0.67 | 0.71 | UK | 0.57 | 0.66 |
| EN | 0.53 | 0.61 | SV | 0.57 | 0.70 |
| ES | 0.53 | 0.61 | PT | 0.54 | 0.65 |
| FA | 0.52 | 0.61 | ZH | 0.58 | 0.63 |
| FR | 0.56 | 0.64 | | | |
| HI | 0.71 | 0.77 | MULTI | 0.63 | 0.69 |

Table 3: XLMR performance on MULTICONER V2.

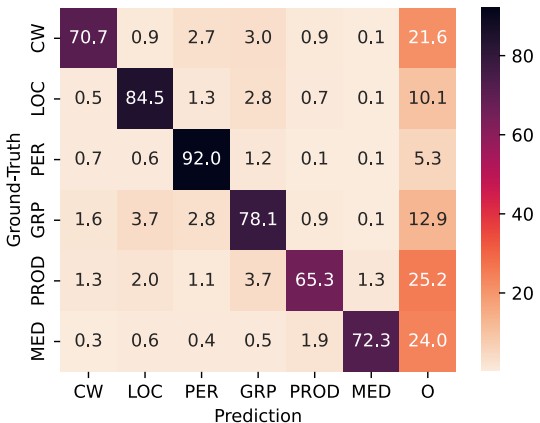

Figure 3: Coarse grained errors on the MULTI test set.

## 3.2 Impact of Fine-Grained Taxonomy

Performance varies greatly across fine-grained types within a coarse type (Table 10). Using MULTI as a representative baseline, we note a

gap of up to 34% in F1 scores, as seen with CREATIVEWORK and PERSON, indicating the challenges of introducing fine-grained types not seen at the coarse level (Malmasi et al., 2022b).

The gap can be attributed to two primary components: (i) the incorrect classification of entities under inappropriate sub-categories or broad categories, and (ii) recall problems where certain entities are missed (assigned the O tag).

**Misclassifications** Figure 3 shows the distribution of predicted coarse tags for MULTI. Most of the misclassifications are for the GROUP class, which is often misclassified with LOCATION or PERSON. Yet, this does not reveal how the models handle the specific classes within a coarse type.

To understand the challenges presented by our taxonomy for NER models, we further inspect misclassifications within the coarse type GROUP in Figure 7 of Appendix D. There are several types that are misclassified, e.g., PUBLICCORP as ORG, or PRIVATECORP with ORG or PUBLICCORP. On the other hand, MUSICALGRP is often misclassified with types from other coarse types, such as ARTIST or VISUALWORK. This highlights some of the issues and challenges that NER models face when exposed to more fine-grained types.

**Recall Issues** Figure 3 shows the rate of entities marked with the O tag. Consistent recall issues affect the MEDICAL, CREATIVEWORK, and PRODUCT classes, often with more than 20% of entities being missed. Across languages the recall can vary, with even higher portions of missed entities (e.g., FA, ZH, EN with more than 30+% of MEDICAL entities missed, cf. Figure 6).

## 3.3 Impact of Corruption Strategies

Figure 4 shows the impact of noise level on the macro-F1 score, where we see a strong negative correlation between noise level and performance (Spearman's correlation across different languages is $\widehat{\rho} = -0.998$). For ZH, the impact is higher than for the rest of languages and the macro-F1 decreases faster: with 50% of test data corrupted, the score decreases approximately by 9%, while for the rest it drops only between 4% and 5.5%.

Table 4 shows the impact of corruption strategies on *context* and *entity* tokens, targeting exclusively the respective tokens. For both variants, the number of corruptions per sampled with $\lambda = 1$

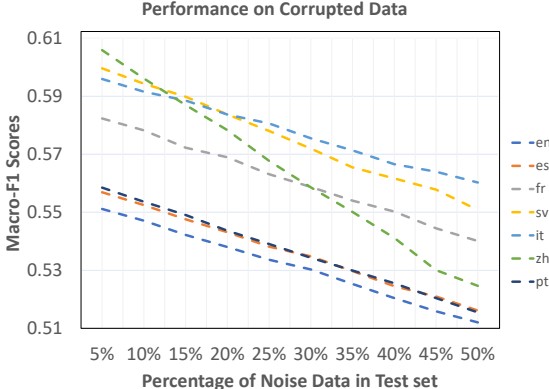

Figure 4: Noise impact on performance. X-axis shows the rate of noise, and Y-axis shows the macro-F1 score. Our final test set applies noise to 30% of sentences.

| Corruption Target | EN | ES | FR | IT | PT | SV | UK | ZH |
|---|---|---|---|---|---|---|---|---|
| Context | 0.48 | 0.51 | 0.50 | 0.54 | 0.51 | 0.55 | 0.54 | 0.55 |
| Entity | 0.42 | 0.42 | 0.45 | 0.46 | 0.42 | 0.44 | 0.41 | 0.48 |

Table 4: macro-F1 scores when all the test data is corrupted on *context* or *entity* tokens.

and $\delta \leq 4$ hyper-parameters. Results show that entity token corruption has a notably detrimental effect on performance, in contrast to context token corruption. On average across all languages, the macro-F1 score for corrupted entities is approximately 9% lower than the performance on corrupted context, emphasizing the higher impact of entity noise compared to context noise. Finally, Table 11 shows the baseline's performance breakdown on the clean and noisy subsets of the test set.

## 4 Conclusions and Future Work

We presented MULTICONER V2, a new large-scale dataset that represents a number of current challenges in NER. The results from XLMR indicate that our dataset is difficult, with a macro-F1 score of only 0.63 on the MULTI test set.

The results presented illustrate that the MULTICONER V2, with its detailed NER taxonomy and noisy test sets, poses substantial challenges to large pre-trained language models' performance. It is our hope that this resource will help further research for building better NER systems. This dataset can serve as a benchmark for evaluating NER methods that infuse external entity knowledge, as well as assessing NER model robustness on noisy scenarios.

We envision several directions for future work. The extension of MULTICONER V2 to additional languages is the most straightforward direction for future work. Expansion to other more challenging domains, such as search queries (Fetahu et al., 2021), can help explore fine-grained NER within even shorter inputs.

Finally, evaluating the performance of this data using zero-shot and few-shot cross-lingual models (Fetahu et al., 2022) or even unsupervised NER models (Iovine et al., 2022) can help us better understand the role of data quantity in building effective NER systems.

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

# Appendix

## A  NER Challenges & Motivation

Table 5 lists the key challenges that are introduced in MULTICONER V2 for NER model. The challenges range from low context sentences, complex entities with additionally fine-grained NER taxonomy, to syntactic issues such as lowercasing as well as noisy context and entity tokens.

## B  Dataset Construction

This section provides a detailed description of the methods used to generate our dataset. Namely, how sentences are selected from Wikipedia, and additionally how the sentences are tagged with the corresponding named entities and how the underlying fine-grained taxonomy is constructed. Additionally, the different data splits are described, as well as the license of our dataset.

Table 6 shows the detailed stats about our proposed MULTICONER V2 dataset.

### B.1  Fine-Grained NER Taxonomy

We use our fine-grained NER taxonomy, namely, the entities associated with each type as part of the automated data generation process. To generate entity associations to our fine-grained taxonomy we make use of Wikidata. The entity type graph in Wikidata is noisy, containing *cycles*, and often types are associated with different parent types, thus, making it challenging to use the data as is for generating fine-grained and clean taxonomies for NER. We carry out a series of steps that we explain in the following.

**Type Cycles.** Wikidata type graph contains cycles, which are not suitable for taxonomy generation. We first break the cycles through a depth-first search approach, by following the children of a given type, and whenever a given type has children that are already visited (encountered during the DFS), those edges are cut out.

**Entity to Fine-Grained Type Association:** Our taxonomy consists of 6 coarse types, PERSON, LOCATION, GROUP, MEDICAL, PRODUCT, CREATIVEWORK, and 33 of fine-grained types. The coarse types represent classes that we aim to have in our taxonomy, and with the exception of MEDICAL and PRODUCT, which at the same time

represent contributions and enrichment of the current NER data landscape, represent common types encountered in NER research. We generate the entity to type association at the fine-grained level, given that the association at the coarse level can be inferred automatically.

For each of the fine-grained types we *manually identify* specific Wikidata types, from where we then query the associated entities. Wikidata types and entity type associations are done by the Wikidata community, and thus, often there are redundant entity types (with similar semantics). From the initially identified types and their entities, we expand our set of Wikidata types that are associated to a fine-grained type in our taxonomy, by additionally taking into account other Wikidata types that the entities are associated with. From the identified Wikidata types for a fine-grained type in our taxonomy, we traverse the entire Wikidata type graph, and obtain all the entities that are associated with those types and all of their children.

Finally, this resulted in 16M Wikidata entities, whose distribution across the different coarse and fine-grained types is shown in Table 7 and 8.

**Quality Filtering:** Finally, once we have obtained all the possible Wikidata entities that are associated with the fine-grained types in our NER taxonomy, we need to ensure that the associations are not *ambiguous*.

We tackle the problem ambiguity, namely Wikidata entities being associated with different types, causing them to be in different NER classes at the same time. To resolve such ambiguous cases, we compute the distribution of entities that are associated with a set of fine-grained NER type, and based on a small sample analysis, manually determine with which NER type they should be associated.

### B.2  Sentence Extraction

MULTICONER V2 is constructed from encyclopedic resources such as Wikipedia, with focus on sentences with low context, sampled using heuristics that identify sentences that represent the NER challenges we target in Table 5. Figure 5 shows the steps from our data construction workflow, which for a given language it extracts sentences from the corresponding Wikipedia locale, and applies a series of steps to generate the NER data. This process is performed for the following languages: FA, FR, EN, ES, IT, PT, SV. For the rest of languages

| Challenge | Description |
|---|---|
| **Fine-grained Entities** | The entity type can be different based on the context. For example, a creative work entity "`Harry Potter and the Sorcerer's Stone`" could be s a book or a film, depending on the context. |
| **Noisy NER** | Gazetteer based models would not work for typos (e.g., "`sony xperia`" → "`somy xpria`") or spelling errors (e.g., "`ford cargo`" → "`f0rd cargo`") in entities, degrading significantly their performance. |
| **Ambiguous Entities and Contexts** | Some NEs are ambiguous: they are not always entities, e.g., "`Inside Out`", "`Among Us`", and "`Bonanza`" may refer to NEs (a movie, video game, and TV show) in some contexts, but not in others. Such NEs often resemble regular syntactic constituents. |
| **Surface Features** | Capitalization/punctuation features are large drivers of success in NER (Mayhew et al., 2019), but short inputs (ASR, queries) often lack such surface features. An uncased evaluation is needed to assess model performance. |

Table 5: Challenges addressed by MULTICONER V2.

| Class | Split | EN | DE | FA | FR | ES | UK | SV | HI | BN | ZH | IT | PT | Multi |
|---|---|---|---|---|---|---|---|---|---|---|---|---|---|---|
| | train | 9,294 | 5,508 | 8,006 | 9,295 | 8,360 | 6,441 | 7,695 | 3,609 | 3,778 | 4,862 | 10,387 | 8,241 | 85,476 |
| PER | dev | 481 | 280 | 413 | 483 | 442 | 341 | 445 | 174 | 194 | 239 | 548 | 447 | 4,487 |
| | test | 137,681 | 11,299 | 115,868 | 141,401 | 125,379 | 96,864 | 111,157 | 5,736 | 6,935 | 9,095 | 160,598 | 120,413 | 180,080 |
| | train | 4,084 | 2,466 | 3,661 | 5,438 | 3,606 | 2,907 | 3,714 | 1,646 | 1,981 | 2,264 | 5,048 | 3,839 | 40,654 |
| CW | dev | 215 | 127 | 184 | 268 | 183 | 146 | 200 | 90 | 103 | 112 | 267 | 206 | 2,101 |
| | test | 62,126 | 4,777 | 53,034 | 84,952 | 55,459 | 43,291 | 54,806 | 2,804 | 3,640 | 4,369 | 79,873 | 58,245 | 87,030 |
| | train | 4,224 | 2,815 | 3,209 | 3,745 | 3,632 | 3,204 | 3,459 | 2,273 | 2,227 | 2,696 | 3,416 | 3,788 | 38,688 |
| GRP | dev | 218 | 177 | 180 | 195 | 195 | 151 | 194 | 143 | 122 | 145 | 173 | 200 | 2,093 |
| | test | 60,026 | 4,418 | 38,807 | 52,987 | 50,259 | 39,709 | 46,929 | 3,897 | 3,651 | 4,715 | 46,271 | 48,994 | 73,226 |
| | train | 4,353 | 2,269 | 5,086 | 4,723 | 4,651 | 5,458 | 7,176 | 2,487 | 2,457 | 2,470 | 4,446 | 4,794 | 50,370 |
| LOC | dev | 197 | 117 | 267 | 242 | 230 | 294 | 370 | 133 | 127 | 129 | 248 | 250 | 2,604 |
| | test | 67,901 | 5,306 | 70,907 | 73,373 | 72,996 | 84,643 | 111,879 | 7,172 | 7,375 | 6,170 | 68,564 | 70,923 | 117,257 |
| | train | 1,935 | 1,571 | 2,049 | 1,946 | 1,989 | 2,258 | 1,989 | 1,420 | 1,384 | 1,529 | 1,770 | 1,927 | 21,767 |
| PROD | dev | 109 | 78 | 107 | 100 | 100 | 117 | 112 | 74 | 67 | 73 | 86 | 101 | 1,124 |
| | test | 27,580 | 1,643 | 18,212 | 28,274 | 28,469 | 30,071 | 22,686 | 1,611 | 1,493 | 1,869 | 22,887 | 21,115 | 35,545 |
| | train | 1,559 | 1,322 | 1,651 | 1,230 | 1,669 | 1,688 | 1,381 | 1,435 | 1,396 | 1,407 | 1,376 | 1,850 | 17,964 |
| MED | dev | 76 | 62 | 85 | 64 | 81 | 86 | 70 | 70 | 63 | 75 | 76 | 88 | 896 |
| | test | 22,491 | 1,434 | 15,287 | 17,208 | 23,812 | 20,796 | 13,702 | 1,979 | 1,919 | 1,781 | 19,029 | 21,062 | 29,553 |
| | train | 16,778 | 9,785 | 16,321 | 16,548 | 16,453 | 16,429 | 16,363 | 9,632 | 9,708 | 9,759 | 16,579 | 16,469 | 170,824 |
| **Total** | dev | 871 | 512 | 855 | 857 | 854 | 851 | 856 | 514 | 507 | 506 | 858 | 854 | 8,895 |
| | test | 249,980 | 20,145 | 219,168 | 249,786 | 246,900 | 238,296 | 231,190 | 18,399 | 19,859 | 20,265 | 247,881 | 229,490 | 358,668 |

Table 6: MULTICONER V2 dataset statistics for the different languages for the Train/Dev/Test splits. For each NER class we show the total number of entity instances per class on the different data splits. The bottom three rows show the total number of sentences for each language.

| Coarse Class | #Entities | ratio |
|---|---|---|
| PERSON | 9,586,716 | 57% |
| CREATIVEWORK | 3,146,607 | 18.7% |
| LOCATION | 3,010,448 | 17.9% |
| GROUP | 607,420 | 3.5% |
| PRODUCT | 396,611 | 2.4% |
| MEDICAL | 29,968 | 0.18% |

Table 7: Gazetteer stats at the coarse named entity type level.

(BN, ZH, DE, HI), we apply machine translation to obtain the data, as described below.

In the following we explain the required steps to generate the final output of sentences tagged with the corresponding entity types.

1. **Sentence Extraction:** From the cleaned Wikipedia pages, we extract sentences, which contain hyperlinks that point to entities that can be mapped into their corresponding Wikidata entity equivalent. In this way we ensure that the entities in a sentence are marked with high quality to their corresponding fine-grained types (cf. Table 8).

2. **Sentence Filtering:** We remove sentences that are shorter than 28 characters and longer than 128 characters. In this way, we ensure that the sentences contain complete clauses,[6] and furthermore, by filtering out longer sentences (more than 128 characters) we ensure that the remaining sentences have low context, making NER more challenging. Finally, we additionally filter out sentences that contain capitalized nouns (proper nouns), which are not interlinked

---

[6]i.e., sentences shorter than 28 characters may contain only entity names, and as such are not suitable for NER

| Coarse Class | Fine-Grained Class | #Entities | ratio |
|---|---|---|---|
| | OTHERPER | 4,343,697 | 25.89% |
| | SCIENTIST | 1,901,438 | 11.333% |
| | ARTIST | 1,392,664 | 8.301% |
| PERSON | CLERIC | 134,078 | 0.799% |
| | SPORTSMANAGER | 48,741 | 0.291% |
| | ATHLETE | 1,003,554 | 5.981% |
| | POLITICIAN | 762,544 | 4.545% |
| | FACILITY | 2,470,918 | 14.727% |
| LOCATION | OTHERLOC | 256,644 | 1.53% |
| | STATION | 196,113 | 1.169% |
| | HUMANSETTLEMENT | 86,773 | 0.517% |
| | ORG | 392,390 | 2.339% |
| | MUSICALGRP | 106,643 | 0.636% |
| | SPORTSGRP | 83,902 | 0.5% |
| GROUP | PUBLICCORP | 12,570 | 0.075% |
| | PRIVATECORP | 1,888 | 0.011% |
| | CARMANUFACTURER | 9,528 | 0.057% |
| | AEROSPACEMANUFACTURER | 499 | 0.003% |
| | OTHERPROD | 213,848 | 1.275% |
| | VEHICLE | 144,555 | 0.862% |
| PRODUCT | FOOD | 17,791 | 0.106% |
| | CLOTHING | 11,604 | 0.069% |
| | DRINK | 8,813 | 0.053% |
| | WRITTENWORK | 1,366,197 | 8.143% |
| | ARTWORK | 575,665 | 3.431% |
| CREATIVEWORK | VISUALWORK | 540,773 | 3.223% |
| | MUSICALWORK | 539,878 | 3.218% |
| | SOFTWARE | 87,017 | 0.519% |
| | DISEASE | 14,627 | 0.087% |
| | MEDICATION/VACCINE | 5,947 | 0.035% |
| MEDICAL | MEDICALPROCEDURE | 4,286 | 0.026% |
| | ANATOMICALSTRUCTURE | 4,186 | 0.025% |
| | SYMPTOM | 922 | 0.005% |

Table 8: Gazetteer stats at the coarse named entity type level.

to entities. This is done for languages that follow capitalization rules for proper nouns (e.g. EN and ES). This presents an important step, since if a proper noun is not interlinked to an entity, however it represents one, this will artificially inflate false positive scores for NER models.

### B.3 Sentence Translation

For the languages, for which we cannot automatically extract sentences and their corresponding tagged entities (e.g. DE uses capitalization for all nouns, thus, we cannot accurately distinguish nouns against proper nouns), we apply automatic translation to generate two portions of our data. For four languages (BN, DE, HI, ZH), Wikipedia sentences are translated from the EN data.

We use the Google Translation API[7] to perform our translations. The input texts may contain known entity spans or slots.

To prevent the spans from being translated, we leveraged the notranslate attribute to mark them and prevent from being translated. The translation quality in the different languages such as

BN, ZH, DE and HI is very high, with over 90% translation accuracy (i.e., accuracy as measured by human annotators in terms of the translated sentence retaining the semantic meaning and as well have a correct syntactic structure in the target language).

### B.4 Data Splits

To ensure that obtained NER results on this dataset are *reproducible*, we create three predefined sets for training, development and testing. The entities in each class are distributed following a root square normalization, which takes into account the actual distribution of a given entity class. Additionally we set a minimum number of entities from a class that need to appear in each split. This combination allows us not to heavily bias the data splits towards the more popular classes (e.g. WRITTENWORK). Table 6 shows detailed statistics for each of the 13 subtasks and data splits.

**Training Data** For the training data split, we limit the size to approx. 16k sentences. The number of instances was chosen to be comparable to well-known NER datasets such as CoNLL03 (Sang and De Meulder, 2003). Note that in the case of the Multi subset, the training split contains all the instances from the individual language splits.

**Development Data** We randomly sample around ∼800 instances,[8] a reasonable amount of data for assessing model generalizability.

**Test Data** Finally, the testing set represents the remaining instances that are not part of the training or development set. To avoid exceedingly large test sets, we limit the number of instances in the test set to be around at most 250k sentences (cf. Table 6). The only exception is for the Multi, which was generated from the language specific test splits, and was downsampled to contain at most 35k from each monolingual test set, resulting in a total of 358k instances. The larger test set sizes are ideal to assess the generalizability and robustness of models on unseen entities.

### B.5 License, Availability, and File Format

The dataset is released under a CC BY-SA 4.0 license, which allows adapting the data. Details

---

[7]https://cloud.google.com/translate

[8]With the exception of the translated languages, DE, HI, BN, and ZH, where the data is scarce.

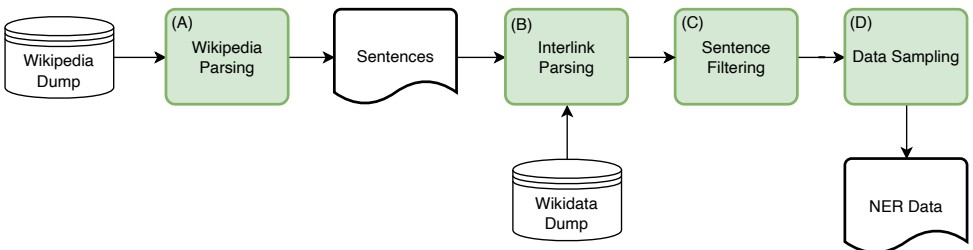

Figure 5: An overview of the different steps involved in extracting the MULTICONER V2 data from Wikipedia dumps, similar to MULTICONER.

about the license are available on the Creative Commons website.[9]

The data is distributed using the commonly used BIO tagging scheme in CoNLL03 format (Sang and De Meulder, 2003). The complete dataset will be available for download.[1]

### B.6 Noise Generation

| Original Character | Visually Similar Ones | Phonetically Similar Ones |
|---|---|---|
| 干 | 千,午,汗 | 肝,敢,甘 |
| 防 | 肪,访,妨 | 方,房,芳 |

Table 9: Examples of visually and phonetically similar ZH characters.

Finally, as part of the data construction, we generate a subset of corrupted test set (30%) that adds noise to either the context or entity tokens as described in in §2.3. The purpose of the noisy test set is to assess the robustness of NER models, trained on clean data, in spotting correctly the named entities that may contain either context or entity noise. The noise mimics typing noise or noise that may be attributed to OCR, where characters are misrecognized with visually similar characters.

For all the languages the noise is applied at the token level, where random characters within a token are replaced with either a character that comes from typing errors (neighboring characters on a given keyboard layout for a specific language) or visually similar characters. For ZH is applied at character level, this is due to the language structure. Table 9 shows examples of visually similar and phonetically similar characters in ZH.

Figure 2 shows examples of corrupted sentences from seven different languages.

### C  Fine Grained NER Performance

**Performance on Fine-grained Classes:** Table 10 shows the results for the different fine-grained classes and languages. We note a high divergence in terms of performance for the different fine-grained types within a coarse type.

**Performance on Noise and Clean Test Subsets:** From Table 11, we can see that the simulated noise can have a huge impact on the performance of the baseline model. On average, the Macro-F1 score on noisy subset is about 10% lower than that on the clean subset. In this section, we study the impact of different corruption strategies, and amount of corruption on the test set. Note that, differently from the fixed amount of noise of 30% applied on the test set, here we assess the range of noise with up to 50% of the test data.

### D  Fine Grained NER Error Analysis

Here we show the errors that the XLMR baseline makes at the fine-grained type level. Figure 7 shows the misclassifications of the baseline approach. Here, we notice that within the GROUP type there are several fine-grained types that are often confused with other types within GROUP, e.g. PUBLICCORP often misclassified as ORG, or PRIVATECORP often misclassified with ORG or PUBLICCORP. On the other hand, MUSICALGRP is often misclassified with other fine-grained types from other coarse types, such as ARTIST or VISUALWORK.

The error analysis highlights some of the issues that NER models face when exposed to more fine-grained types. Furthermore, through this analysis, we provide insights on the challenges presented by the newly proposed NER taxonomy and areas of improvement in terms of NER performance.

---

[9] https://creativecommons.org/licenses/by-sa/4.0

| | | DE | EN | ES | FR | SV | IT | ZH | PT | FA | BN | HI | UK | MULTI |
|---|---|---|---|---|---|---|---|---|---|---|---|---|---|---|
| MED | ANATOMICALSTRUCTURE | 0.79 | 0.59 | 0.59 | 0.52 | 0.64 | 0.60 | 0.63 | 0.58 | 0.47 | 0.77 | 0.81 | 0.67 | 0.67 |
| | DISEASE | 0.81 | 0.61 | 0.62 | 0.59 | 0.65 | 0.56 | 0.66 | 0.63 | 0.54 | 0.82 | 0.85 | 0.66 | 0.68 |
| | MEDICATION/VACCINE | 0.81 | 0.67 | 0.65 | 0.65 | 0.67 | 0.65 | 0.62 | 0.68 | 0.64 | 0.81 | 0.81 | 0.74 | 0.73 |
| | MEDICALPROCEDURE | 0.77 | 0.53 | 0.54 | 0.55 | 0.55 | 0.58 | 0.58 | 0.57 | 0.52 | 0.79 | 0.84 | 0.54 | 0.63 |
| | SYMPTOM | 0.55 | 0.40 | 0.37 | 0.49 | 0.40 | 0.41 | 0.41 | 0.32 | 0.44 | 0.79 | 0.83 | 0.46 | 0.55 |
| PER | ARTIST | 0.74 | 0.72 | 0.69 | 0.76 | 0.72 | 0.81 | 0.68 | 0.72 | 0.72 | 0.69 | 0.71 | 0.69 | 0.76 |
| | ATHLETE | 0.75 | 0.73 | 0.68 | 0.70 | 0.66 | 0.83 | 0.72 | 0.64 | 0.49 | 0.71 | 0.77 | 0.76 | 0.74 |
| | CLERIC | 0.56 | 0.44 | 0.50 | 0.54 | 0.50 | 0.61 | 0.39 | 0.55 | 0.42 | 0.71 | 0.65 | 0.54 | 0.56 |
| | POLITICIAN | 0.58 | 0.49 | 0.50 | 0.54 | 0.59 | 0.48 | 0.52 | 0.54 | 0.53 | 0.64 | 0.69 | 0.49 | 0.57 |
| | SCIENTIST | 0.41 | 0.37 | 0.34 | 0.42 | 0.34 | 0.40 | 0.46 | 0.31 | 0.29 | 0.55 | 0.31 | 0.42 | 0.42 |
| | SPORTSMANAGER | 0.58 | 0.49 | 0.54 | 0.51 | 0.36 | 0.63 | 0.54 | 0.47 | 0.48 | 0.65 | 0.25 | 0.57 | 0.58 |
| | OTHERPER | 0.48 | 0.41 | 0.40 | 0.43 | 0.47 | 0.42 | 0.44 | 0.45 | 0.38 | 0.52 | 0.54 | 0.44 | 0.48 |
| LOC | FACILITY | 0.69 | 0.58 | 0.53 | 0.60 | 0.67 | 0.66 | 0.60 | 0.57 | 0.53 | 0.70 | 0.70 | 0.57 | 0.64 |
| | HUMANSETTLEMENT | 0.88 | 0.83 | 0.79 | 0.78 | 0.91 | 0.82 | 0.75 | 0.83 | 0.75 | 0.86 | 0.88 | 0.85 | 0.84 |
| | OTHERLOC | 0.54 | 0.45 | 0.38 | 0.43 | 0.88 | 0.43 | 0.57 | 0.65 | 0.30 | 0.81 | 0.75 | 0.55 | 0.57 |
| | STATION | 0.78 | 0.69 | 0.66 | 0.71 | 0.73 | 0.66 | 0.82 | 0.68 | 0.75 | 0.87 | 0.83 | 0.70 | 0.73 |
| PROD | FOOD | 0.67 | 0.47 | 0.50 | 0.49 | 0.58 | 0.48 | 0.57 | 0.53 | 0.52 | 0.65 | 0.75 | 0.55 | 0.57 |
| | DRINK | 0.66 | 0.45 | 0.55 | 0.50 | 0.59 | 0.53 | 0.43 | 0.58 | 0.48 | 0.82 | 0.87 | 0.56 | 0.60 |
| | SOFTWARE | 0.77 | 0.59 | 0.72 | 0.64 | 0.70 | 0.68 | 0.55 | 0.70 | 0.60 | 0.82 | 0.82 | 0.73 | 0.74 |
| | VEHICLE | 0.68 | 0.42 | 0.48 | 0.42 | 0.54 | 0.48 | 0.59 | 0.43 | 0.49 | 0.68 | 0.76 | 0.55 | 0.54 |
| | CLOTHING | 0.46 | 0.49 | 0.40 | 0.48 | 0.49 | 0.37 | 0.48 | 0.36 | 0.27 | 0.34 | 0.84 | 0.44 | 0.51 |
| | OTHERPROD | 0.62 | 0.40 | 0.44 | 0.49 | 0.58 | 0.48 | 0.52 | 0.58 | 0.53 | 0.58 | 0.65 | 0.51 | 0.57 |
| GRP | MUSICALGRP | 0.73 | 0.57 | 0.63 | 0.64 | 0.70 | 0.76 | 0.61 | 0.67 | 0.60 | 0.68 | 0.87 | 0.74 | 0.72 |
| | PRIVATECORP | 0.64 | 0.19 | 0.33 | 0.38 | 0.14 | 0.36 | 0.70 | 0.00 | 0.26 | 0.95 | 0.91 | 0.13 | 0.54 |
| | CARMANUFACTURER | 0.69 | 0.51 | 0.59 | 0.57 | 0.55 | 0.62 | 0.59 | 0.52 | 0.64 | 0.70 | 0.76 | 0.65 | 0.63 |
| | AEROSPACEMANUFACTURER | 0.69 | 0.41 | 0.32 | 0.44 | 0.16 | 0.21 | 0.56 | 0.16 | 0.62 | 0.08 | 0.17 | 0.26 | 0.54 |
| | PUBLICCORP | 0.61 | 0.49 | 0.61 | 0.55 | 0.53 | 0.59 | 0.51 | 0.70 | 0.55 | 0.70 | 0.80 | 0.67 | 0.67 |
| | ORG | 0.67 | 0.52 | 0.54 | 0.52 | 0.59 | 0.52 | 0.61 | 0.58 | 0.55 | 0.82 | 0.84 | 0.62 | 0.61 |
| | SPORTSGRP | 0.86 | 0.73 | 0.70 | 0.71 | 0.78 | 0.75 | 0.78 | 0.74 | 0.82 | 0.90 | 0.94 | 0.81 | 0.78 |
| CW | ARTWORK | 0.46 | 0.38 | 0.25 | 0.37 | 0.25 | 0.50 | 0.46 | 0.07 | 0.10 | 0.06 | 0.12 | 0.22 | 0.43 |
| | VISUALWORK | 0.73 | 0.61 | 0.62 | 0.79 | 0.71 | 0.87 | 0.63 | 0.64 | 0.73 | 0.71 | 0.74 | 0.69 | 0.77 |
| | WRITTENWORK | 0.76 | 0.58 | 0.59 | 0.66 | 0.61 | 0.56 | 0.64 | 0.56 | 0.48 | 0.74 | 0.76 | 0.60 | 0.64 |
| | MUSICALWORK | 0.77 | 0.67 | 0.59 | 0.59 | 0.67 | 0.76 | 0.54 | 0.67 | 0.53 | 0.61 | 0.71 | 0.55 | 0.70 |
| | macro-F1 | 0.67 | 0.53 | 0.53 | 0.56 | 0.57 | 0.58 | 0.58 | 0.54 | 0.52 | 0.68 | 0.72 | 0.57 | 0.63 |
| | micro-F1 | 0.71 | 0.61 | 0.61 | 0.64 | 0.70 | 0.70 | 0.63 | 0.65 | 0.61 | 0.74 | 0.77 | 0.66 | 0.69 |

Table 10: XLM-RoBERTa baseline results on the MULTICONER V2 test set as measured by the F1 score for the different NER tags.

| | | EN | | ES | | FR | | SV | | IT | | ZH | | PT | |
|---|---|---|---|---|---|---|---|---|---|---|---|---|---|---|---|
| | | clean | noisy | clean | noisy | clean | noisy | clean | noisy | clean | noisy | clean | noisy | clean | noisy |
| MED | ANATOMICALSTRUCTURE | 0.61 | 0.55 | 0.62 | 0.51 | 0.54 | 0.47 | 0.67 | 0.57 | 0.62 | 0.56 | 0.70 | 0.44 | 0.60 | 0.53 |
| | DISEASE | 0.63 | 0.55 | 0.65 | 0.54 | 0.60 | 0.55 | 0.69 | 0.57 | 0.57 | 0.53 | 0.73 | 0.50 | 0.65 | 0.60 |
| | MEDICATION/VACCINE | 0.70 | 0.62 | 0.68 | 0.58 | 0.68 | 0.56 | 0.71 | 0.57 | 0.68 | 0.57 | 0.67 | 0.54 | 0.72 | 0.61 |
| | MEDICALPROCEDURE | 0.55 | 0.49 | 0.57 | 0.47 | 0.58 | 0.48 | 0.60 | 0.42 | 0.61 | 0.52 | 0.65 | 0.44 | 0.59 | 0.53 |
| | SYMPTOM | 0.43 | 0.32 | 0.41 | 0.26 | 0.54 | 0.35 | 0.45 | 0.30 | 0.42 | 0.37 | 0.52 | 0.23 | 0.35 | 0.22 |
| PER | ARTIST | 0.74 | 0.69 | 0.70 | 0.65 | 0.77 | 0.73 | 0.73 | 0.69 | 0.82 | 0.78 | 0.73 | 0.58 | 0.73 | 0.68 |
| | ATHLETE | 0.74 | 0.70 | 0.69 | 0.66 | 0.71 | 0.66 | 0.67 | 0.64 | 0.84 | 0.80 | 0.76 | 0.63 | 0.66 | 0.61 |
| | CLERIC | 0.46 | 0.39 | 0.52 | 0.47 | 0.55 | 0.51 | 0.51 | 0.47 | 0.63 | 0.58 | 0.46 | 0.26 | 0.58 | 0.48 |
| | POLITICIAN | 0.50 | 0.46 | 0.52 | 0.46 | 0.55 | 0.51 | 0.60 | 0.56 | 0.50 | 0.45 | 0.57 | 0.42 | 0.56 | 0.50 |
| | SCIENTIST | 0.37 | 0.37 | 0.35 | 0.31 | 0.43 | 0.41 | 0.35 | 0.31 | 0.41 | 0.38 | 0.50 | 0.35 | 0.32 | 0.27 |
| | SPORTSMANAGER | 0.50 | 0.46 | 0.55 | 0.53 | 0.52 | 0.48 | 0.36 | 0.35 | 0.64 | 0.61 | 0.62 | 0.36 | 0.48 | 0.44 |
| | OTHERPER | 0.43 | 0.38 | 0.41 | 0.37 | 0.45 | 0.40 | 0.49 | 0.44 | 0.43 | 0.40 | 0.49 | 0.33 | 0.46 | 0.43 |
| LOC | FACILITY | 0.62 | 0.50 | 0.57 | 0.45 | 0.63 | 0.53 | 0.71 | 0.57 | 0.69 | 0.60 | 0.65 | 0.51 | 0.60 | 0.49 |
| | HUMANSETTLEMENT | 0.85 | 0.77 | 0.82 | 0.73 | 0.81 | 0.72 | 0.93 | 0.87 | 0.85 | 0.76 | 0.80 | 0.62 | 0.85 | 0.77 |
| | STATION | 0.73 | 0.61 | 0.68 | 0.59 | 0.75 | 0.62 | 0.78 | 0.60 | 0.70 | 0.58 | 0.88 | 0.67 | 0.71 | 0.60 |
| | OTHERLOC | 0.49 | 0.36 | 0.41 | 0.29 | 0.48 | 0.32 | 0.96 | 0.65 | 0.47 | 0.32 | 0.60 | 0.51 | 0.69 | 0.55 |
| PROD | FOOD | 0.49 | 0.43 | 0.53 | 0.42 | 0.51 | 0.45 | 0.62 | 0.49 | 0.49 | 0.44 | 0.66 | 0.38 | 0.56 | 0.44 |
| | DRINK | 0.48 | 0.38 | 0.60 | 0.43 | 0.55 | 0.37 | 0.63 | 0.48 | 0.58 | 0.41 | 0.52 | 0.31 | 0.64 | 0.38 |
| | SOFTWARE | 0.62 | 0.53 | 0.76 | 0.63 | 0.68 | 0.57 | 0.74 | 0.61 | 0.70 | 0.61 | 0.59 | 0.48 | 0.74 | 0.61 |
| | VEHICLE | 0.44 | 0.36 | 0.49 | 0.43 | 0.44 | 0.37 | 0.57 | 0.45 | 0.51 | 0.43 | 0.68 | 0.36 | 0.47 | 0.33 |
| | CLOTHING | 0.51 | 0.44 | 0.43 | 0.34 | 0.50 | 0.43 | 0.53 | 0.39 | 0.40 | 0.29 | 0.61 | 0.30 | 0.42 | 0.23 |
| | OTHERPROD | 0.42 | 0.36 | 0.47 | 0.38 | 0.51 | 0.43 | 0.61 | 0.50 | 0.50 | 0.44 | 0.58 | 0.41 | 0.61 | 0.52 |
| GRP | MUSICALGRP | 0.60 | 0.50 | 0.66 | 0.55 | 0.68 | 0.57 | 0.73 | 0.63 | 0.78 | 0.71 | 0.67 | 0.49 | 0.69 | 0.61 |
| | PRIVATECORP | 0.20 | 0.16 | 0.31 | 0.36 | 0.40 | 0.32 | 0.11 | 0.19 | 0.36 | 0.33 | 0.77 | 0.48 | 0.00 | 0.00 |
| | CARMANUFACTURER | 0.54 | 0.42 | 0.63 | 0.51 | 0.61 | 0.47 | 0.58 | 0.48 | 0.66 | 0.52 | 0.65 | 0.51 | 0.54 | 0.46 |
| | AEROSPACEMANUFACTURER | 0.45 | 0.32 | 0.37 | 0.22 | 0.48 | 0.36 | 0.18 | 0.12 | 0.23 | 0.17 | 0.00 | 0.00 | 0.20 | 0.09 |
| | PUBLICCORP | 0.52 | 0.41 | 0.65 | 0.51 | 0.58 | 0.48 | 0.57 | 0.45 | 0.63 | 0.51 | 0.60 | 0.40 | 0.73 | 0.63 |
| | ORG | 0.57 | 0.42 | 0.58 | 0.45 | 0.57 | 0.44 | 0.64 | 0.49 | 0.55 | 0.45 | 0.68 | 0.50 | 0.62 | 0.48 |
| | SPORTSGRP | 0.77 | 0.66 | 0.74 | 0.62 | 0.76 | 0.61 | 0.81 | 0.70 | 0.77 | 0.69 | 0.84 | 0.67 | 0.77 | 0.68 |
| CW | ARTWORK | 0.39 | 0.37 | 0.26 | 0.22 | 0.39 | 0.31 | 0.28 | 0.19 | 0.54 | 0.41 | 0.00 | 0.08 | 0.07 | 0.05 |
| | VISUALWORK | 0.63 | 0.55 | 0.65 | 0.57 | 0.81 | 0.75 | 0.74 | 0.64 | 0.89 | 0.84 | 0.67 | 0.55 | 0.67 | 0.59 |
| | WRITTENWORK | 0.61 | 0.51 | 0.62 | 0.53 | 0.70 | 0.57 | 0.65 | 0.52 | 0.58 | 0.50 | 0.69 | 0.55 | 0.59 | 0.47 |
| | MUSICALWORK | 0.69 | 0.63 | 0.61 | 0.53 | 0.62 | 0.55 | 0.70 | 0.62 | 0.78 | 0.72 | 0.57 | 0.49 | 0.69 | 0.61 |

Table 11: XLM-RoBERTa baseline results on the MULTICONER V2 test set as measured by the *macro* F1 score for the different NER tags.

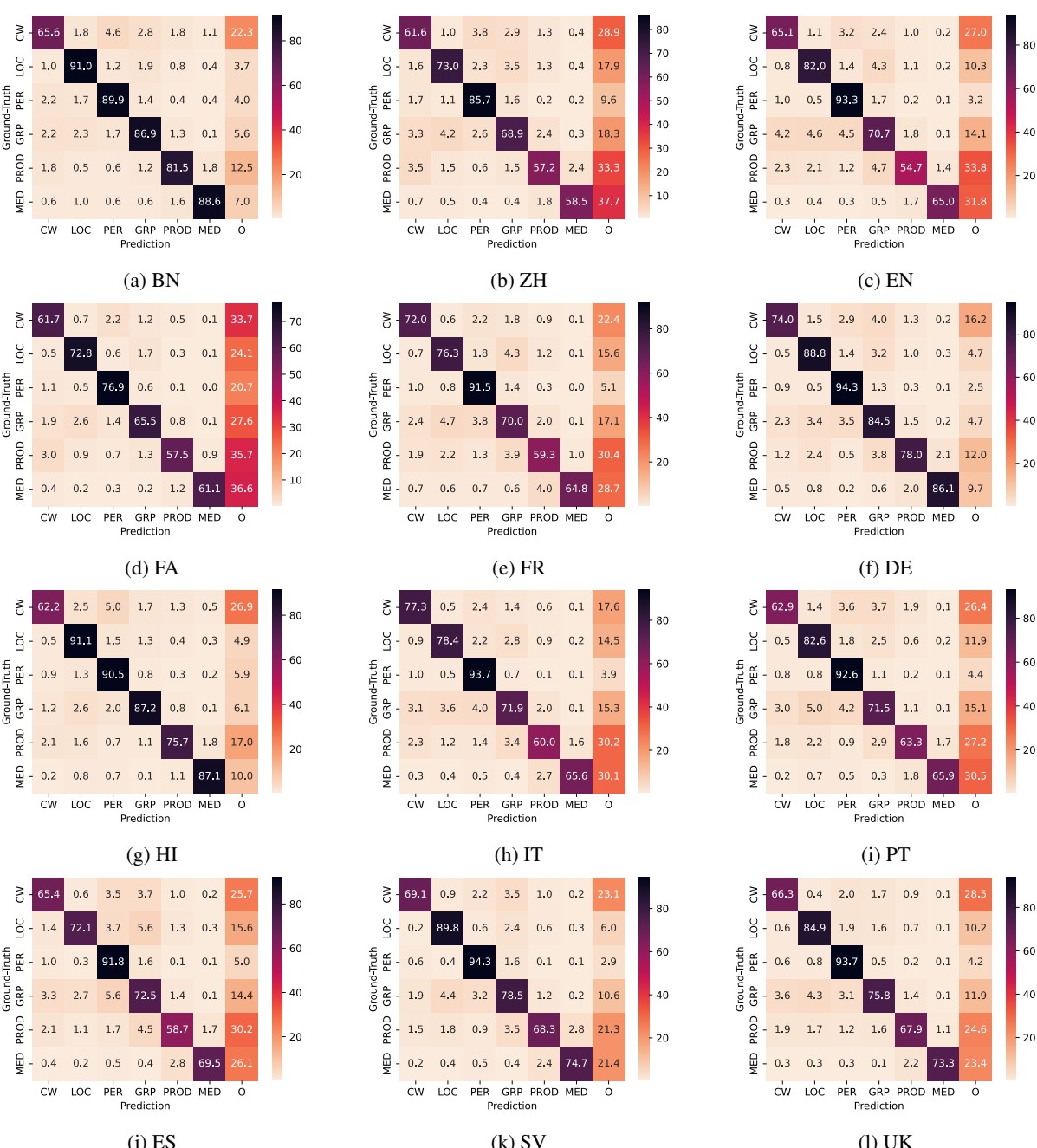

Figure 6: Error analysis for the baseline approach. The plots on the x-axis show the predict coarse class, while the y-axis shows the ground-truth class. The scores represent the ratio of entities of a specific class and its predicted class. The diagonal represent correct class predictions.

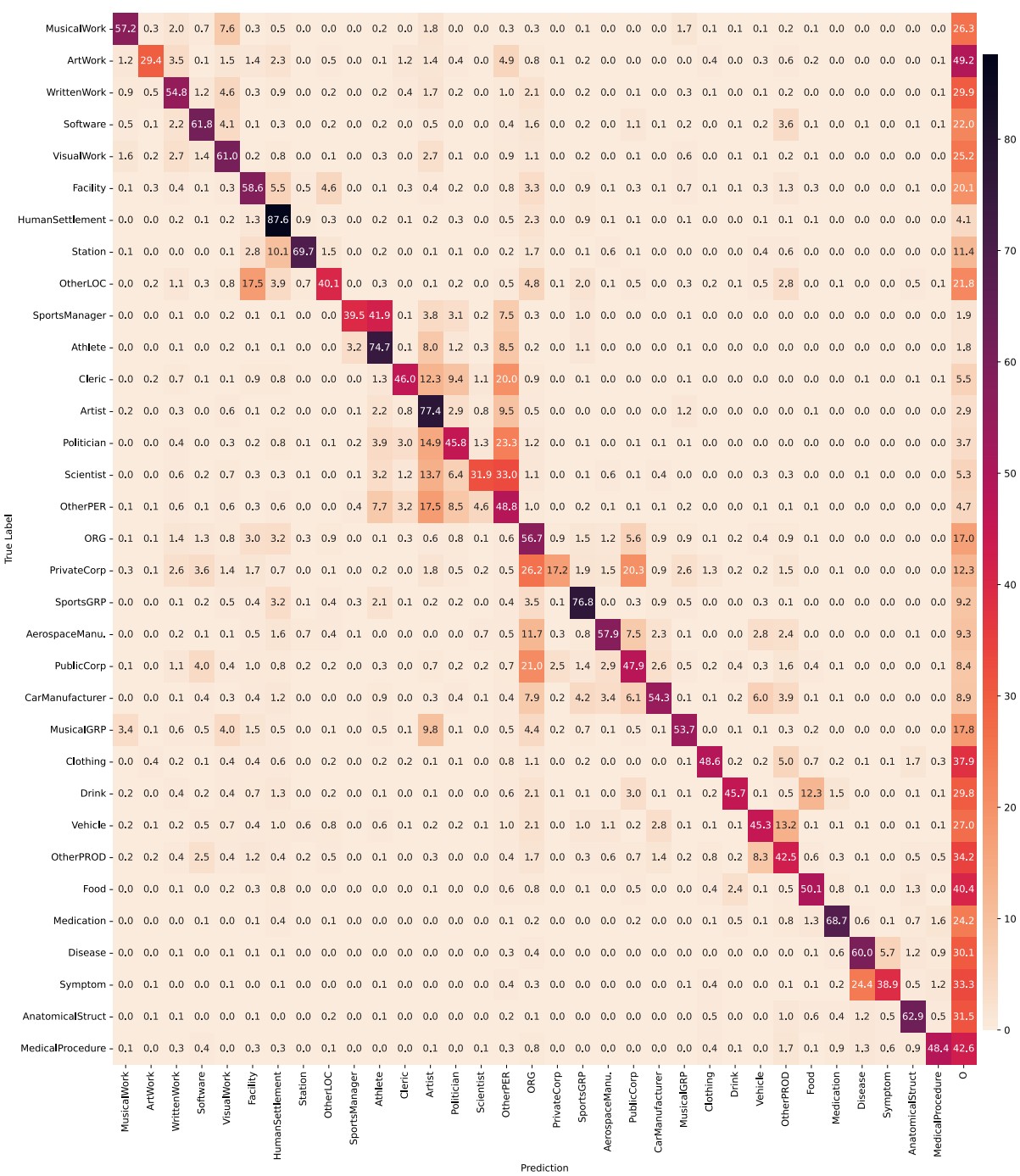

Figure 7: Fine-grained NER predictions for the XLM-RoBERTa baseline on the EN test set.

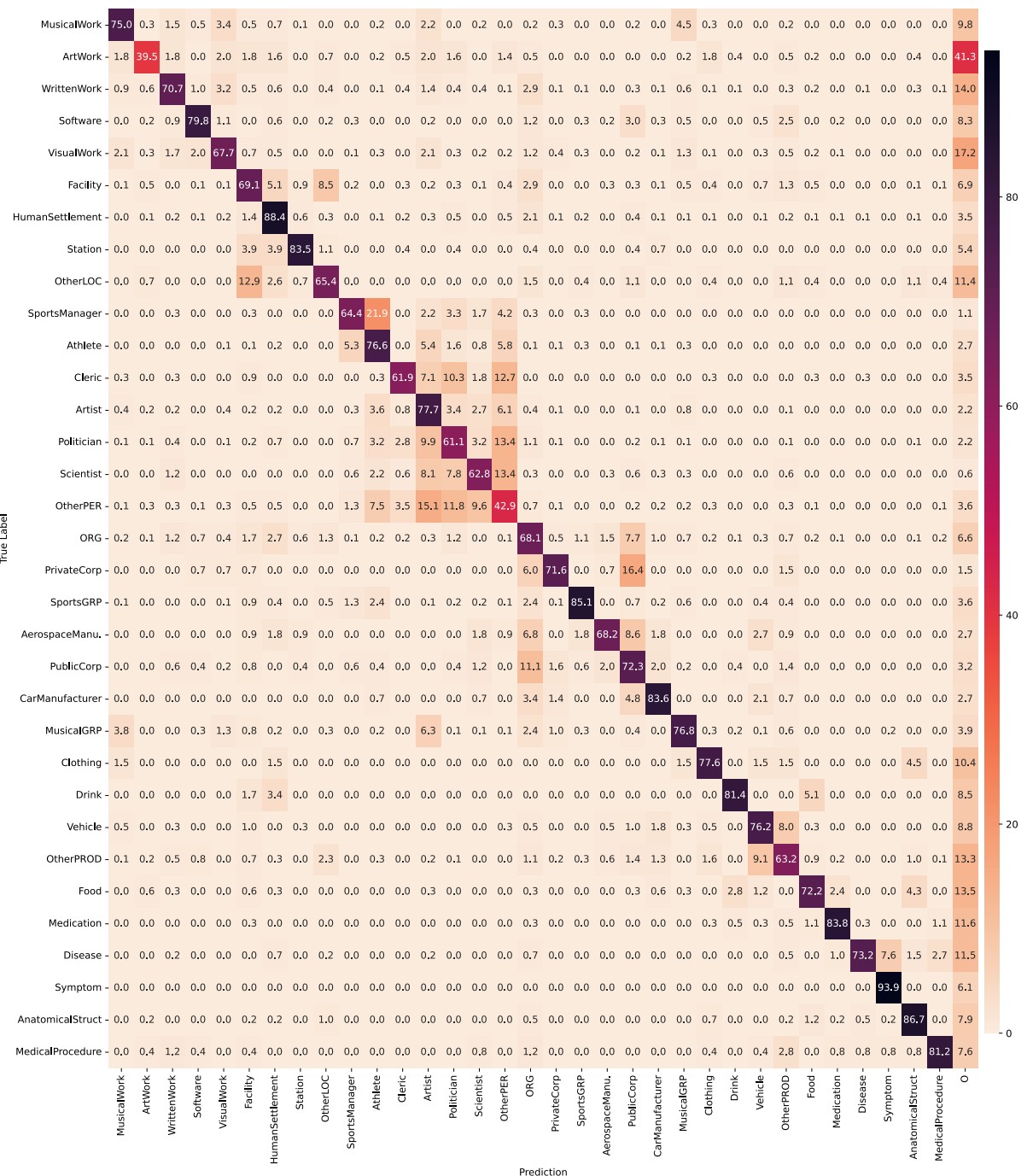

Figure 8: Fine-grained NER predictions for the XLM-RoBERTa baseline on the DE test set.

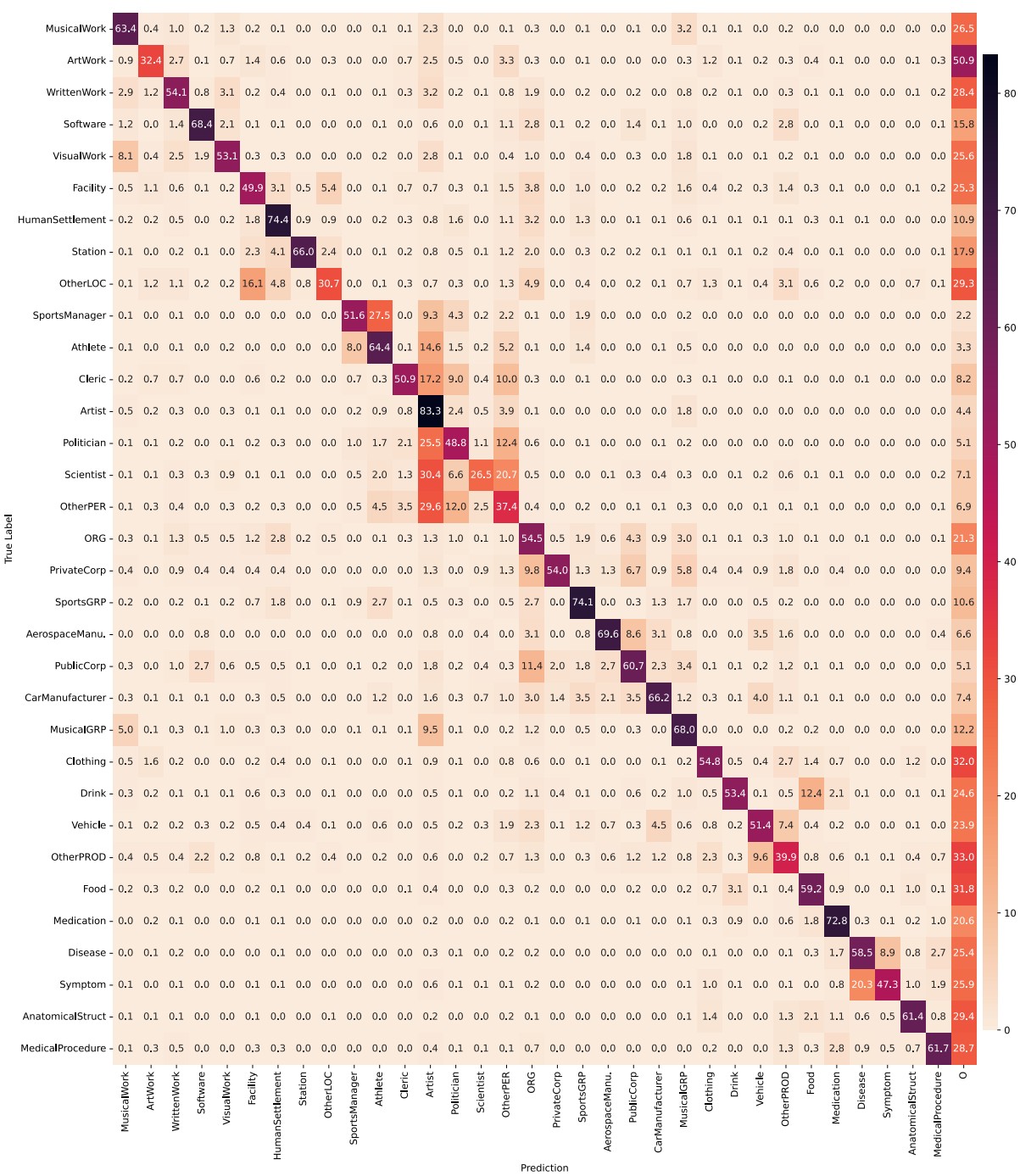

Figure 9: Fine-grained NER predictions for the XLM-RoBERTa baseline on the `ES` test set.

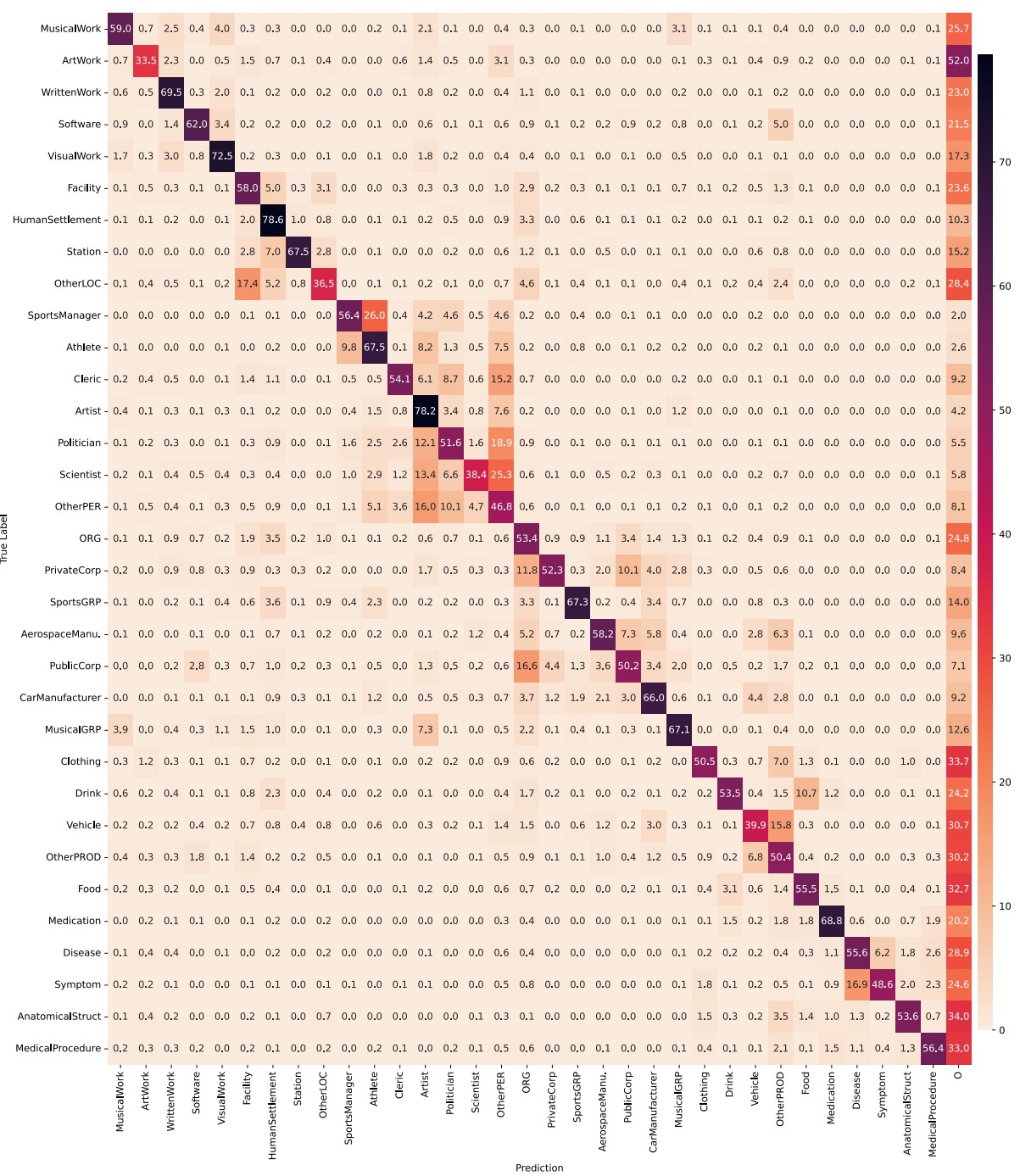

Figure 10: Fine-grained NER predictions for the XLM-RoBERTa baseline on the FR test set.

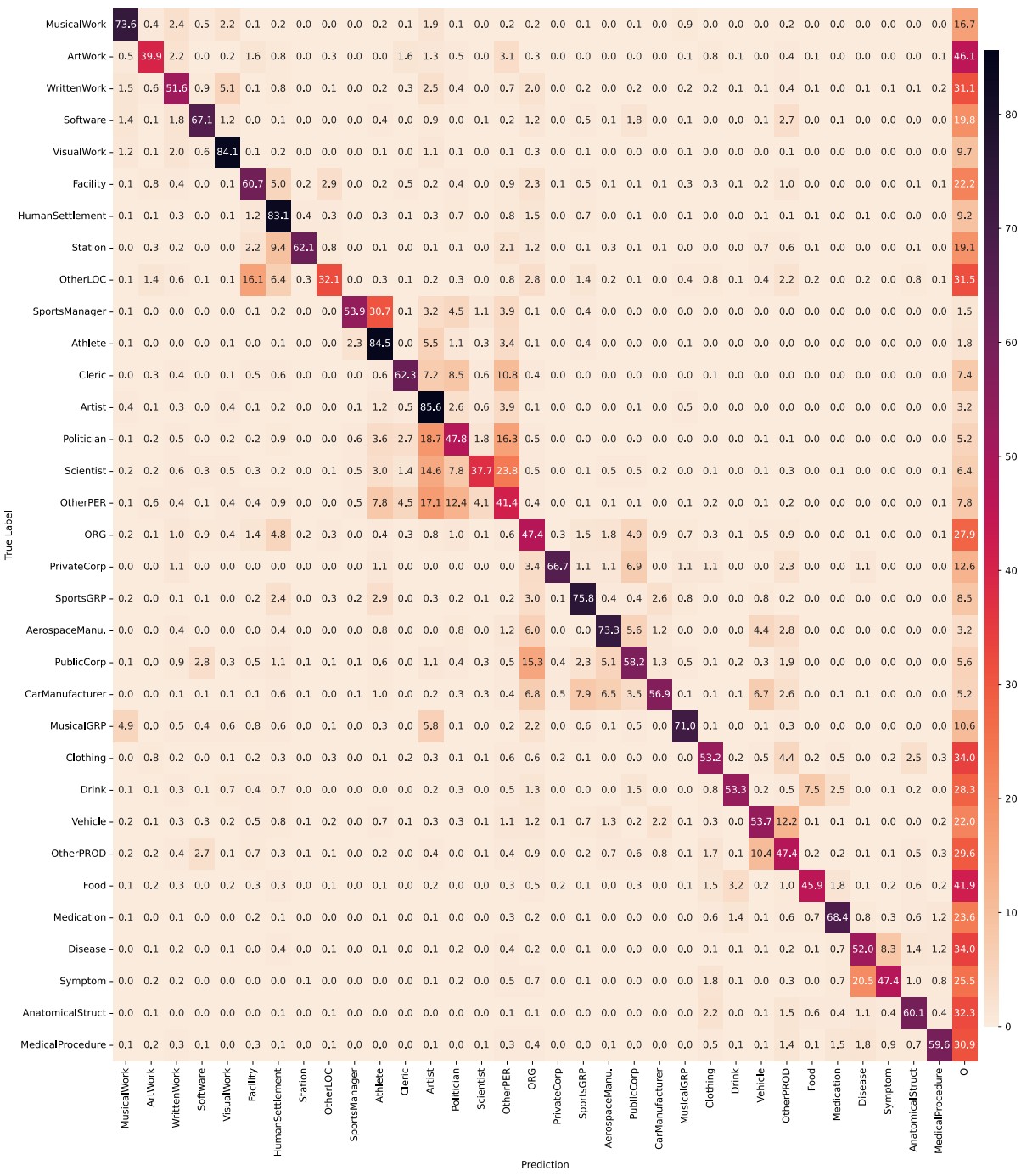

Figure 11: Fine-grained NER predictions for the XLM-RoBERTa baseline on the IT test set.

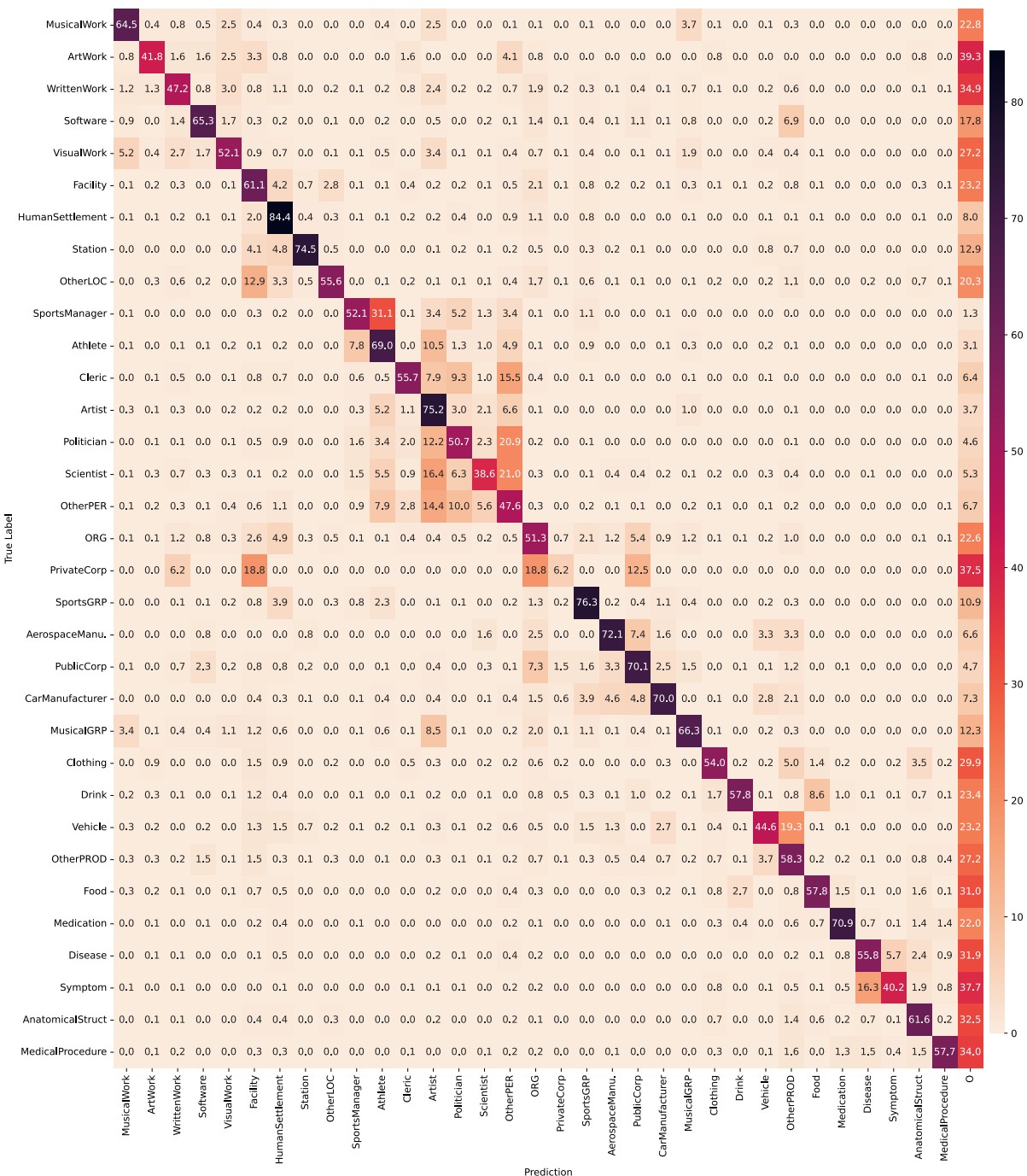

Figure 12: Fine-grained NER predictions for the XLM-RoBERTa baseline on the `PT` test set.

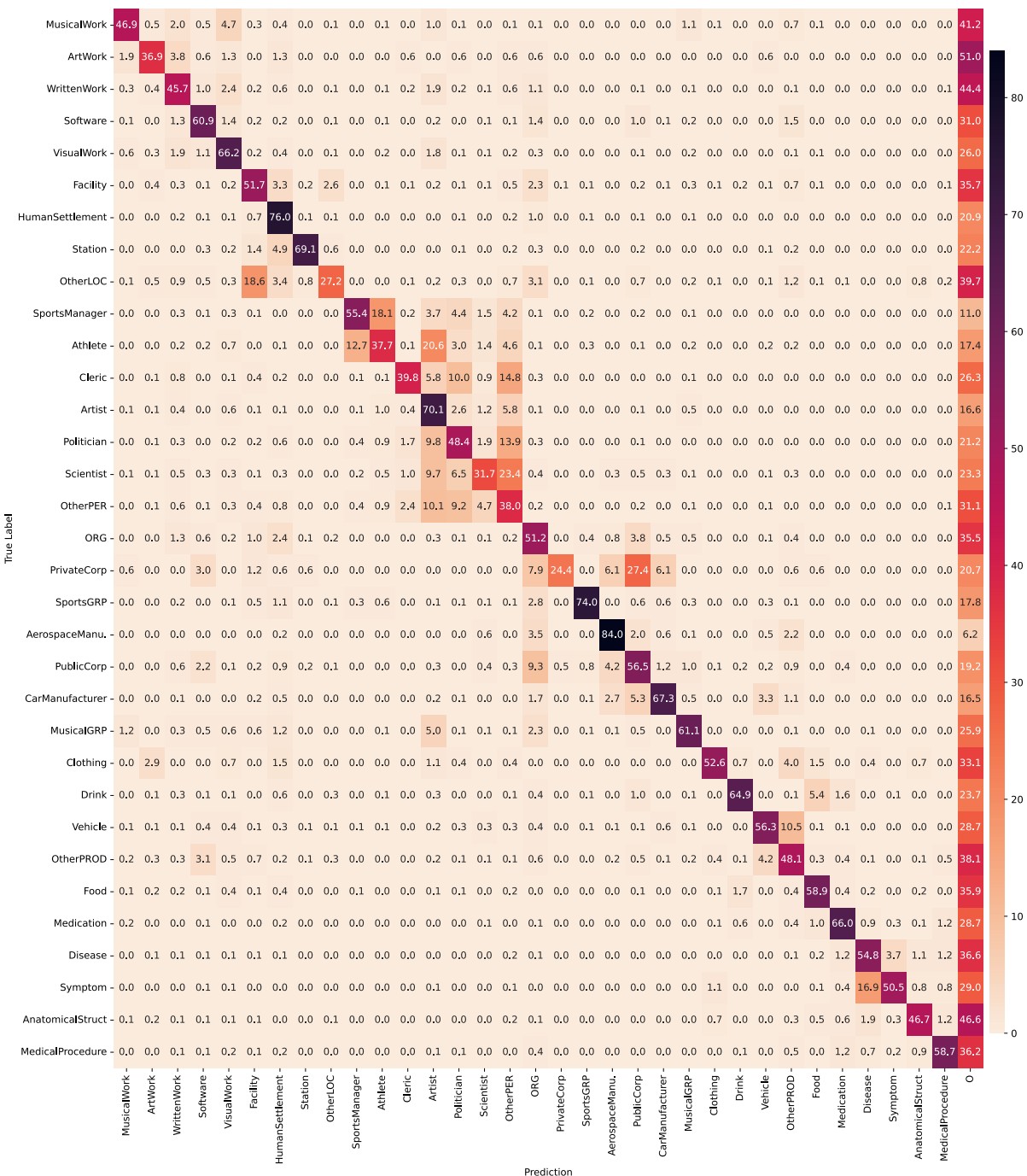

Figure 13: Fine-grained NER predictions for the XLM-RoBERTa baseline on the `FA` test set.

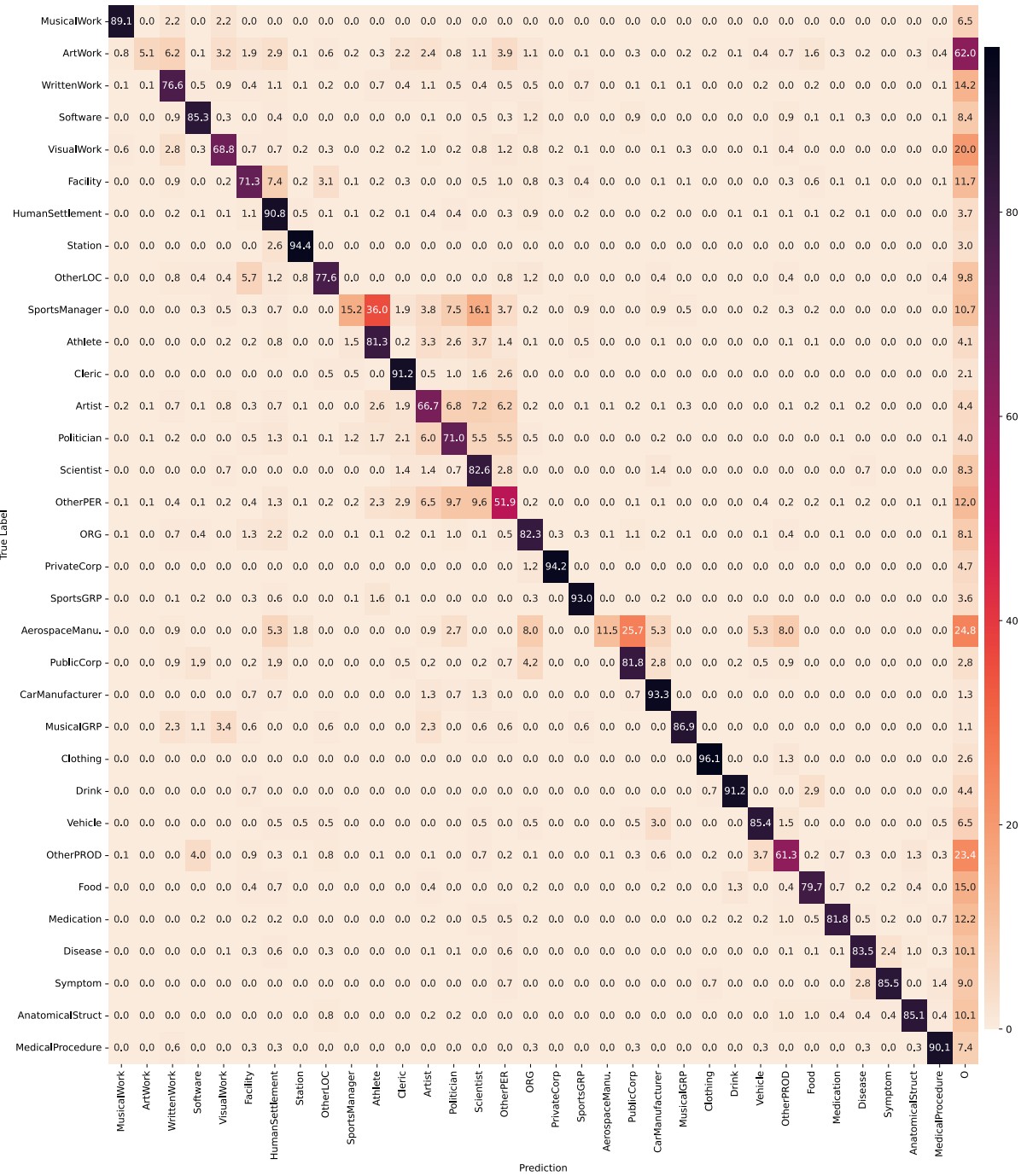

Figure 14: Fine-grained NER predictions for the XLM-RoBERTa baseline on the `HI` test set.

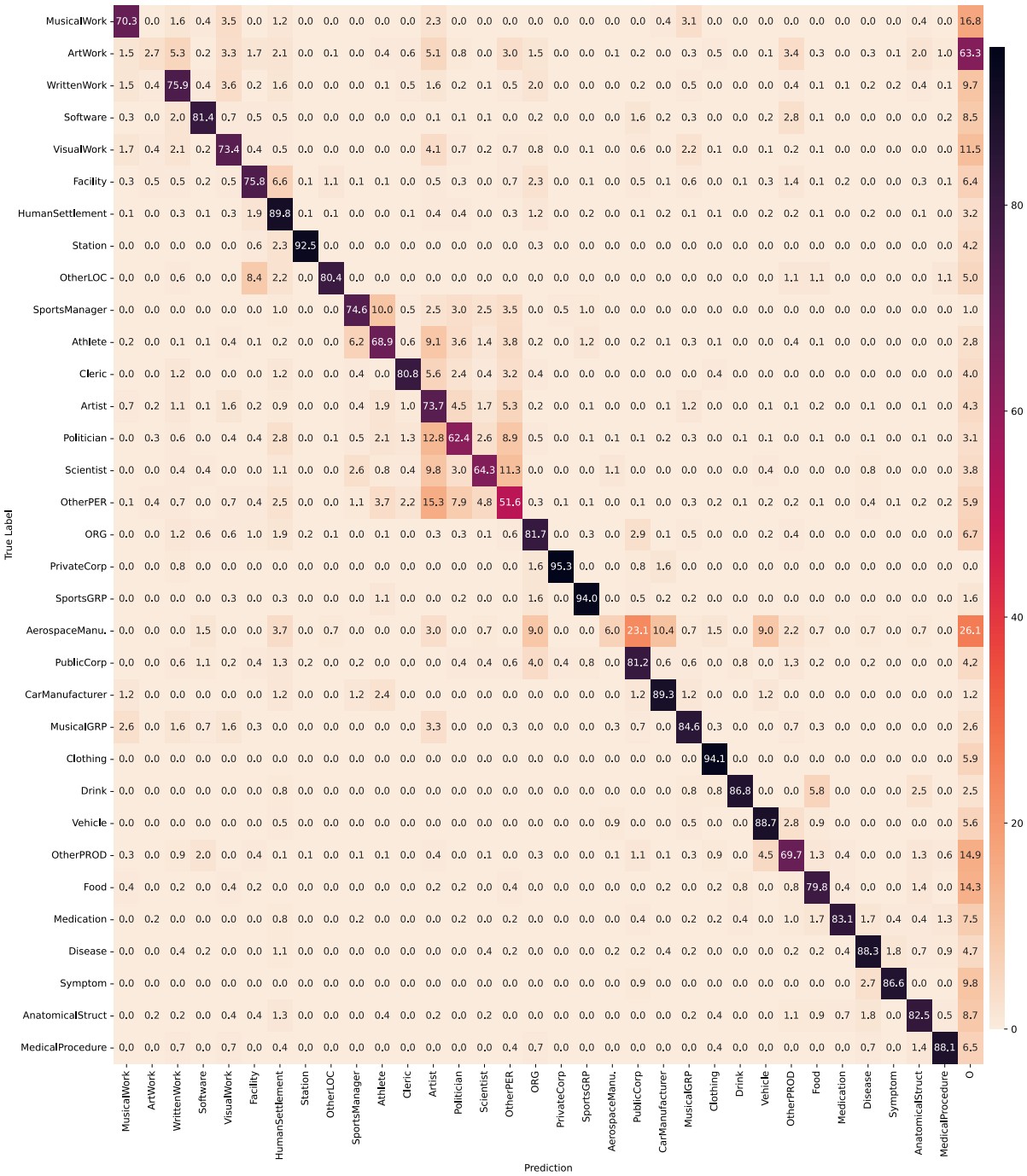

Figure 15: Fine-grained NER predictions for the XLM-RoBERTa baseline on the BN test set.

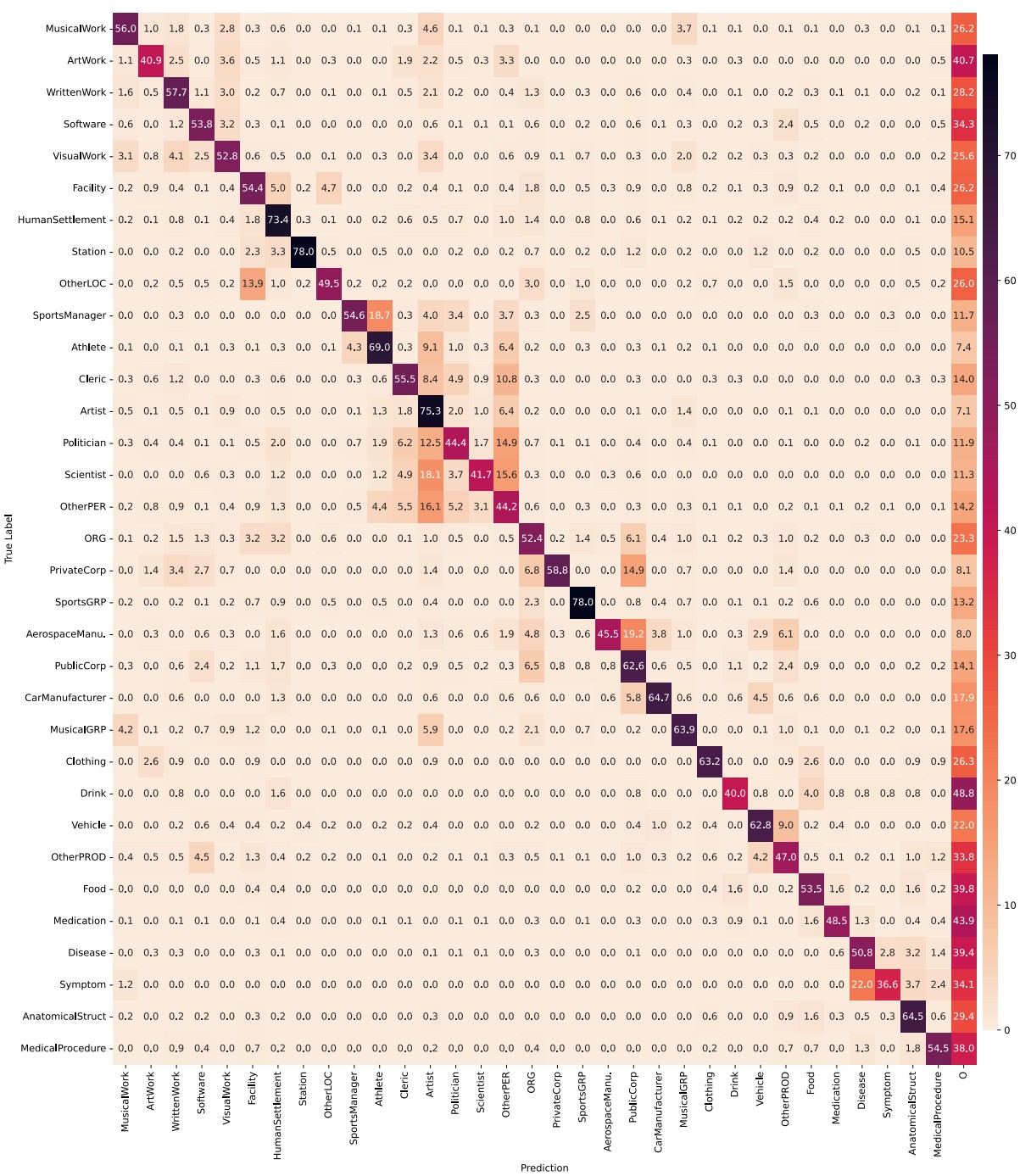

Figure 16: Fine-grained NER predictions for the XLM-RoBERTa baseline on the ZH test set.

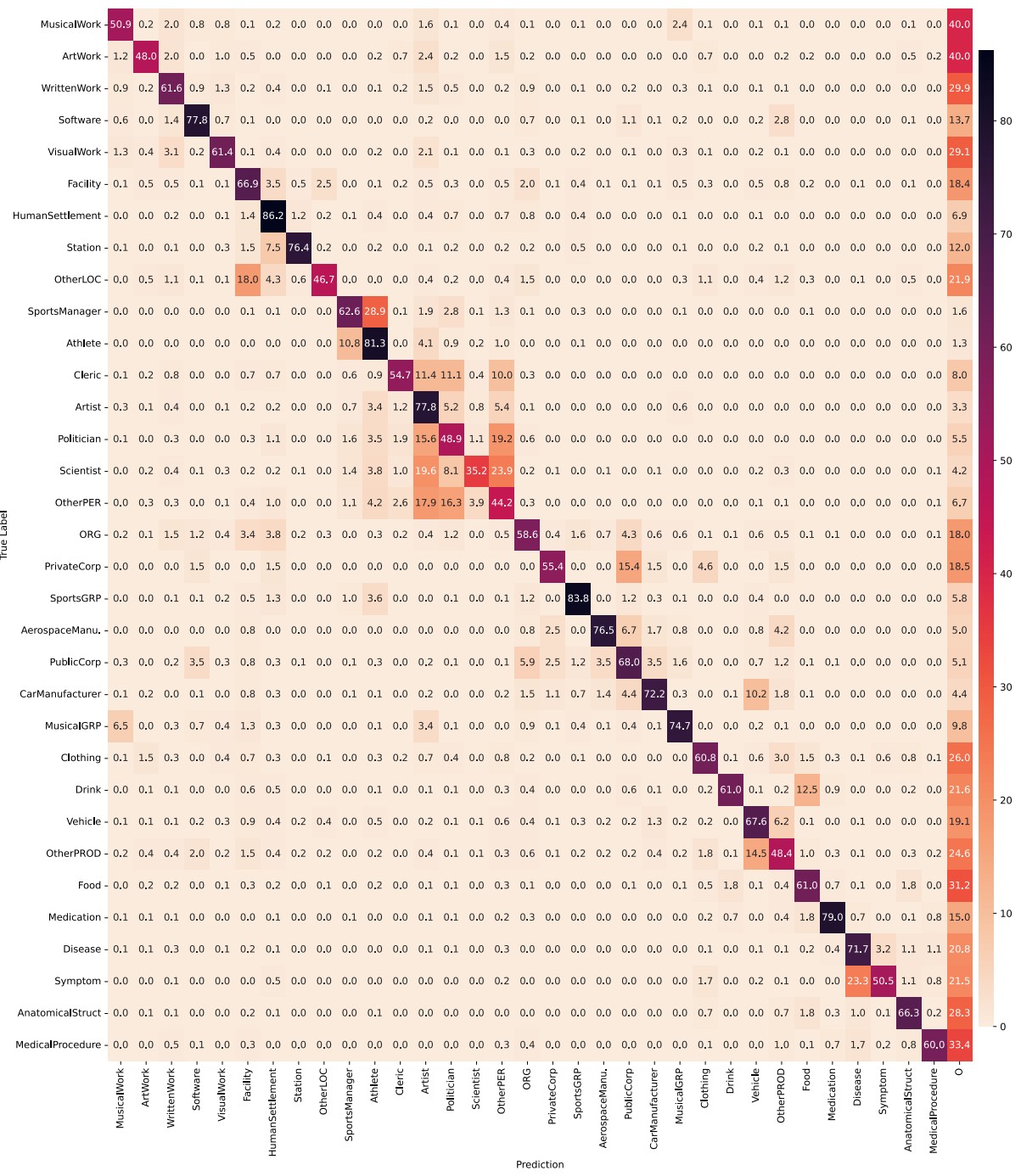

Figure 17: Fine-grained NER predictions for the XLM-RoBERTa baseline on the UK test set.

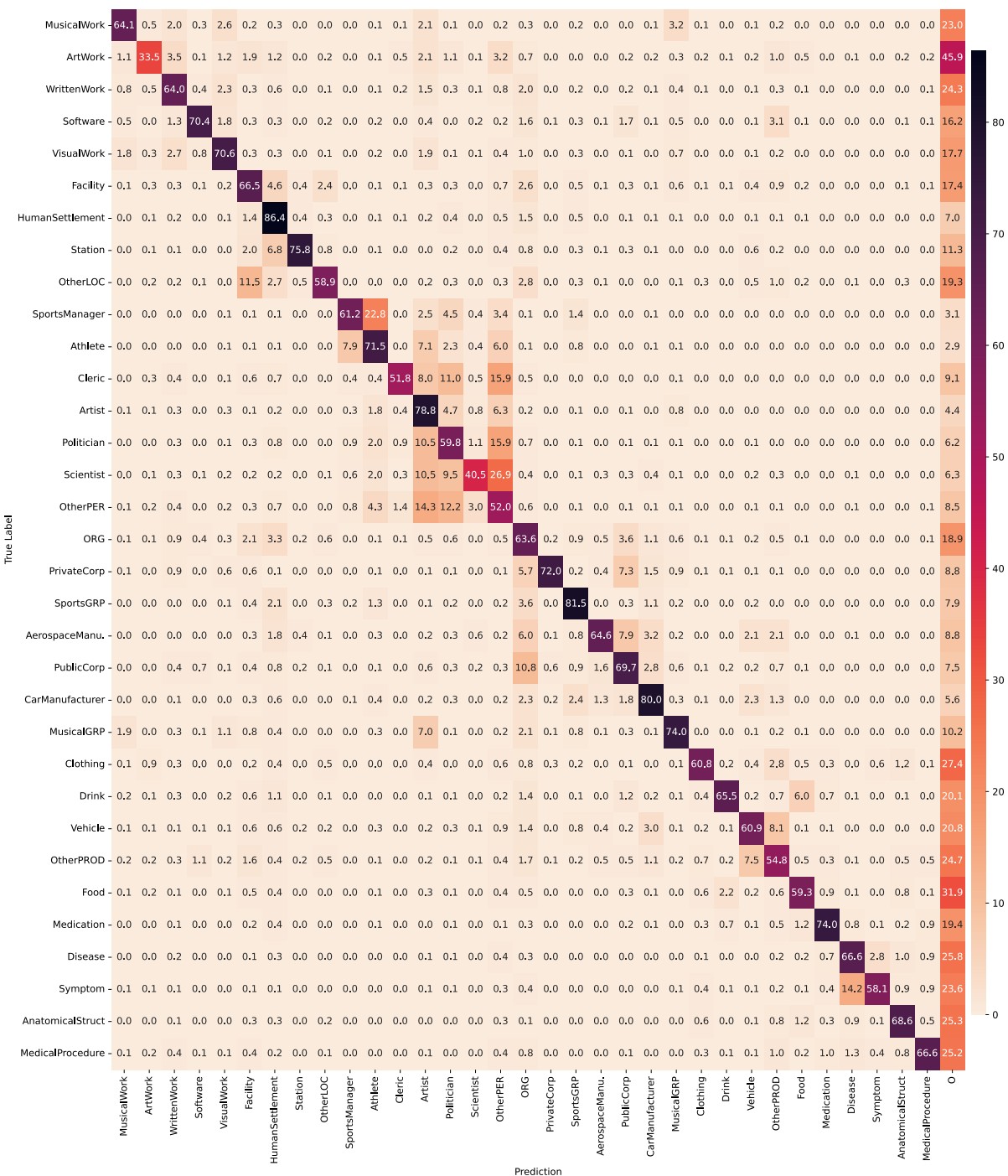

Figure 18: Fine-grained NER predictions for the XLM-RoBERTa baseline on the `MULTI` test set.