# OpenReview forum: "MultiCoNER v2: a Large Multilingual dataset for Fine-grained and Noisy Named Entity Recognition"
_EMNLP/2023/Conference — EMNLP 2023 Findings_

### Official Review · Reviewer_SHVo · 2023-07-31

**Soundness:** 3

**Excitement:**

4: Strong: This paper deepens the understanding of some phenomenon or lowers the barriers to an existing research direction.

**Paper Topic And Main Contributions:**

This authors introduce a new dataset, AnonNER, for fine-grained Named Entity Recognition (NER) in 12 languages, with some of the partes of the dataset created though translation of the Englihs one (Bangla, Chinese, German, Hindi). This dataset aims to address two key challenges in NER: effective context representation from short texts to differentiate between 33 fine-grained entity types in complex scenarios, and performance degradation due to noise such as typing mistakes or OCR errors. The dataset is compiled from public resources like Wikipedia and Wikidata.

The process of building the dataset is identical to MultiCoNER (which indicates that ANON- means anonymized).

**Reasons To Accept:**

- The dataset approaches realistic challenges such as the size of the textual content (e.g., which can be short textual snippets like Web
queries), the characteristics of the entities to be extracted (i.e., capitalization/punctuation features that are missing and the fine-granularity), unseen and emerging entities (e.g., entities that have a fast growth rate such as new songs and movies are released weekly), and the noise that can result from the digitisation process of documents.

- The dataset covers a diverse typology and writing systems, and range from well-resourced (e.g., EN) to low-resourced ones, which is an important contribution.

**Reasons To Reject:**

- The paper was submitted as a short paper, however, most of the interesting details are in the Appendix - an astonishing amount of 20 pages! - (such as dataset creation details, use of translation for some languages, sentence tokenization, etc.)
- While the contribution is important for the information extraction community, the releasing of such a dataset could be more fit into a conference such as LREC, where the authors could take advantage of the maximum page length for including the actual process of the dataset creation.

**Reproducibility:**

5: Could easily reproduce the results.

**Reviewer Confidence:**

3: Pretty sure, but there's a chance I missed something. Although I have a good feel for this area in general, I did not carefully check the paper's details, e.g., the math, experimental design, or novelty.

**Typos Grammar Style And Presentation Improvements:**

- the tables do not respect the ACL template, the captions are differently positioned, too close to the tables, seems more like it could come from a package clash.

---

> ### Author Rebuttal · Authors · 2023-08-28
>
> We thank the reviewer for their constructive and insightful feedback on our paper.
>
> 1) As pointed to the response for the reviewer iV2c, we wanted to have in the main body what we believed to be the most complete set of steps that were necessary for generating the dataset. The appendix, as pointed out, is in rich in examples, detailed explanations, and results, where we wanted to show the usefulness of this created resource.
>
>    Considering EMNLP’s call for papers, and aspects associated with the Resource track, we believe that given that the paper is self-contained in the main body, the appendix should be considered as a positive aspect rather a reason to reject, since it only shows the utility of the provided resource. Detailed explanations on resource creation are very common to be part of appendices when published at conferences.
>
> 2) We thank the reviewer for their suggestion and we will take it under consideration. However, we are still convinced that our dataset paper is highly suited for EMNLP’s resource track.

---

### Official Review · Reviewer_iV2c · 2023-08-07

**Soundness:** 3

**Excitement:**

4: Strong: This paper deepens the understanding of some phenomenon or lowers the barriers to an existing research direction.

**Missing References:**

https://proceedings.neurips.cc/paper_files/paper/2022/hash/09723c9f291f6056fd1885081859c186-Abstract-Datasets_and_Benchmarks.html
https://xiaoling.github.io/figer/
https://aclanthology.org/C18-1060/

**Paper Topic And Main Contributions:**

The authors propose a new dataset for fine-grained multi-lingual NER dataset for which also includes noisy inputs mimicking visual and phonetic errors at the character levels. They run extensive analysis using XLMR model and provide a strong baseline for the dataset along with the impact of different types of noise on evaluation scores.

**Reasons To Accept:**

Well written paper, good dataset, especially liked the section on evaluation based on noise.
Supplementary section provides a lot of details.


**Reasons To Reject:**

Fine-grained NER is a well studied problem, so the novelty of this work is less. I would have loved to see improvement in performance on a regular benchmark by using this dataset which would make the data more valuable.
Dataset details should be part of the main paper even if done in a brief manner as that is the main contribution of the paper.
No link to the data provided with the paper.
Some missing references around noisy and fine-grained NER datasets.

**Reproducibility:**

4: Could mostly reproduce the results, but there may be some variation because of sample variance or minor variations in their interpretation of the protocol or method.

**Reviewer Confidence:**

5: Positive that my evaluation is correct. I read the paper very carefully and I am very familiar with related work.

---

> ### Author Rebuttal · Authors · 2023-08-28
>
> We thank the reviewer for their highly constructive and insightful feedback on our paper.
>
> 1) In terms of benchmarking, this dataset is the first to address the following aspects: 1) coverage of diverse languages both in terms of script (Latin, Cyrillic, Arabic, Farsi, Chinese, Hindi, Bangla etc.), 2) mix of low and high resources, and 3) scope of named entities to be detected as part of NER, both in terms of fine-grained NER and as well as other aspects such as noisy entities. To the best of our understanding this is the first work to contribute at this scale and problems at once.
>
> 2) We will add more details in the main body of the paper. However, given the nature of Resource track papers, where a lot technical aspects must be taken into account, we have limited ourselves to incorporate in the main body what we determined to be the complete process of the dataset creation. The appendix is rich in additional explanations, examples, results, and algorithms, that can help for reproducibility and further extend our proposed dataset creation mechanism for other researchers.
>
> 3) We will incorporate any missing references as suggested by the reviewer.

---

### Official Review · Reviewer_kR1G · 2023-08-09

**Soundness:** 2

**Excitement:**

2: Mediocre: This paper makes marginal contributions (vs non-contemporaneous work), so I would rather not see it in the conference.

**Paper Topic And Main Contributions:**

The authors present a dataset focusing on multilingual fine-grained Named Entity Recognition (NER). This dataset covers a wide range of entity types at varying levels of granularity. The dataset spans a range from English to under-resourced languages like Farsi and Bangla
A unique feature of this dataset is the introduction of texts that have been deliberately modified to introduce lexical noise. This noise is intended to simulate real-world typing errors or mistakes typically seen in OCR processes.
In their evaluation, the authors evaluate XLM-RoBERTa model on their dataset. The results indicate a consistent decline in performance corresponding to the extent of data corruption, for both context and entities.

The main contribution of this submission is the introduction of a noisy test set designed to replicate errors typical of keyboard typing and OCR (specifically, errors stemming from letter similarities). To the best of my knowledge, there is a lack of large-scale datasets addressing this specific niche, and the discourse on how contemporary NER models perform under such conditions is notably absent in the current literature.

**Questions For The Authors:**

A. Could you elaborate on the methodological contributions unique to this dataset that aren't extensively covered in Malmasi et al. (2022a)? Aside from the introduction of the noisy test set, in what ways does the dataset presented here diverge from the aforementioned work?

**Reasons To Accept:**

To my knowledge, large-scale datasets encompassing noisy entities and context are scarce moreover the introduction of typing / letter similarity noise, barring the references cited by the authors. Introducing such a test set could serve as a valuable asset for a segment of the NER community, potentially fostering the development of more robust solutions for NER models.

**Reasons To Reject:**

The framing of the paper's contribution appears to be misleading. This submission primarily presents the construction of a novel multilingual fine-grained NER dataset with the addition of noisy entities and text. Notably, the entire data construction section has been relegated to the appendix, depriving the main body of the paper of crucial information about the dataset and its creation.

Upon a deeper examination of the very long appendix, it is evident that there was significant effort invested in the data construction process, underscored by a detailed pipeline leading to the creation of this large-scale, fine-grained, multilingual dataset. However, the entire process (including some of the plots..) has already been published by Malmasi et al. (2022a) under the title "MultiCoNER", which (in my opinion) stands out as a commendable dataset creation paper.

The authors here neither assert that their dataset is an extension nor a modification of the "MultiCoNER" dataset. Still, they posit the creation of a novel multilingual fine-grained NER dataset with noise. Given that the innovation in the dataset creation seems confined mainly to the introduction of typing/OCR noise, the paper's authentic contribution appears to be minor.
Although the minor contribution might resonate with a segment of the NER and IE community, we should evaluate a paper based on its claims. Given the perceived lack of transparency and the mirroring of an already published methodology, the quality of this paper is diminished. It would be appropriate for the paper to be restructured to accurately reflect its unique contribution.

**Reproducibility:**

3: Could reproduce the results with some difficulty. The settings of parameters are underspecified or subjectively determined; the training/evaluation data are not widely available.

**Reviewer Confidence:**

4: Quite sure. I tried to check the important points carefully. It's unlikely, though conceivable, that I missed something that should affect my ratings.

---

> ### Author Rebuttal · Authors · 2023-08-28
>
> We thank the reviewer for their constructive and insightful feedback on our paper.
>
> In the following we address the reviewer’s criticism regarding novelty and answer their main question.
>
> In comparison to the mentioned MultiCoNER work by Malmasi et al. (2022a), our contributions are two-fold. i) developing a fine-grained taxonomy that will scale to 12 languages, where most of the languages are low-resource, and  developing such dataset for those languages is a complex challenge. While fine-grained NER may not be a novel problem, but for many languages, this is the first fine-grained NER dataset with considerable amount of samples. ii) We have introduced the complex techniques for noise generation in different scenarios like typographical errors, OCR noise. Again, this is not a one-size-fits-all technique. We had to adapt different techniques for different languages. These techniques can scale beyond NER problems alone and can be helpful for low-resource languages.

---

### Meta-Review · Area_Chair_8eZB · 2023-09-10

**Recommendation:** 3

**Metareview:**

This paper present a multilingual corpus for Name Entity Recognition tasks comprising 12 languages which allow for  a diverse typology and writing systems, and range from well-resourced to low-resourced ones.
A fine grained taxonomy is defined leading to 33 entity types.
The objective of this corpus is to evaluate the robustness of NER system (LLM) given the different languages but also under noisy scenario. Word level and character level corruption strategies have been implemented on the test data.
It will be good to clarify the link between this paper and the presented to semeval.

---

### Decision · Program_Chairs · 2023-10-07

**Decision:**

Accept-Findings

**Comment:**

This paper present a multilingual corpus for Name Entity Recognition tasks comprising 12 languages which allow for  a diverse typology and writing systems, and range from well-resourced to low-resourced ones.
A fine grained taxonomy is defined leading to 33 entity types.
The objective of this corpus is to evaluate the robustness of NER system (LLM) given the different languages but also under noisy scenario. Word level and character level corruption strategies have been implemented on the test data.
It will be good to clarify the link between this paper and the presented to semeval.